# Embedding Safety into RL: A New Take on Trust Region Methods

**Nikola Milosevic** [1 2]   **Johannes Müller** [3]   **Nico Scherf** [1 2]

## Abstract

Reinforcement Learning (RL) agents can solve diverse tasks but often exhibit unsafe behavior. Constrained Markov Decision Processes (CMDPs) address this by enforcing safety constraints, yet existing methods either sacrifice reward maximization or allow unsafe training. We introduce Constrained Trust Region Policy Optimization (C-TRPO), which reshapes the policy space geometry to ensure trust regions contain only safe policies, guaranteeing constraint satisfaction throughout training. We analyze its theoretical properties and connections to TRPO, Natural Policy Gradients, and Constrained Policy Optimization. Experiments show that C-TRPO reduces constraint violations while maintaining competitive returns.

## 1. Introduction

Reinforcement Learning (RL) has emerged as a highly successful paradigm in machine learning for solving sequential decision and control problems, with policy gradient (PG) algorithms as a popular approach (Williams, 1992; Sutton et al., 1999; Konda & Tsitsiklis, 1999). Policy gradients are especially appealing for high-dimensional continuous control because they can be easily extended to function approximation. Due to their flexibility and generality, there has been significant progress in enhancing PGs to work robustly with deep neural network-based approaches. PG-based policy optimization methods such as Trust Region Policy Optimization (TRPO) and Proximal Policy Optimization (PPO) are among the most widely used general-purpose reinforcement learning algorithms (Schulman et al., 2017a;b).

While flexibility makes PGs popular among practitioners, it comes at a cost: the policy can explore any behavior during training, posing significant risks in real-world applications. Many methods have been proposed to improve the safety of policy gradients, often based on the Constrained Markov Decision Process (CMDP) framework. However, existing methods either struggle to minimize constraint violations during training or severely limit the agent's performance.

This work introduces a simple strategy to enhance constraint satisfaction in trust region-based safe policy optimization methods without compromising performance. We propose a novel family of policy divergences, inspired by barrier function methods in optimization and safe control, that modify the policy geometry to ensure that trust regions consist only of safe policies. Our approach is motivated by the observation that TRPO and related methods base their trust region on the state-average Kullback-Leibler (KL) divergence. It can be derived as the Bregman divergence induced by the negative conditional entropy on the space of state-action occupancies (Neu et al., 2017).

The key insight of this work is that safer trust regions can be obtained by modifying this function to account for cost constraints. This leads to a provably safe trust region-based policy optimization algorithm that preserves TRPO's guarantees, while simplifying existing methods and reducing constraint violations during training, without sacrificing reward performance.

**Related Work**  Classic solution methods for CMDPs rely on linear programming techniques, see (Altman, 1999). However, they struggle to scale, making them unsuitable for high-dimensional or continuous control problems. While there are numerous scalable approaches to solving CMDPs, we focus on model-free, direct policy optimization methods. Model-based approaches (Berkenkamp et al., 2017; As et al., 2025), are attractive due to their stability and safety guarantees, but require learning a model, which is not always feasible, or imposes additional assumptions on the model space.

*Lagrangian methods* are a widely adopted approach, where the optimization problem is reformulated as a weighted objective that balances rewards and penalties for constraint violations. This is often motivated by Lagrangian duality, where the penalty coefficient is interpreted as the dual variable. Learning the coefficient with stochastic gradient

---

[1]Max Planck Institute for Human Cognitive and Brain Sciences, Leipzig [2]Center for Scalable Data Analytics and Artificial Intelligence (ScaDS.AI), Dresden/Leipzig [3]Institut für Mathematik, Technische Universität Berlin, 10623 Berlin, Germany. Correspondence to: Nikola Milosevic <nmilosevic@cbs.mpg.de>.

*Proceedings of the 42^{nd} International Conference on Machine Learning*, Vancouver, Canada. PMLR 267, 2025. Copyright 2025 by the author(s).

descent presents a popular baseline (Achiam et al., 2017; Ray et al., 2019; Chow et al., 2019; Stooke et al., 2020). However, a naively tuned Lagrange multiplier may not work well in practice due to oscillations and overshoot. To address this issue, (Stooke et al., 2020) uses PID control to tune the dual variable during training, which achieves less oscillation around the constraint and faster convergence to a feasible policy. While Lagrangian approaches are becoming increasingly popular, it is not entirely clear how to update the dual variables during training, which has attracted significant research interest, see e.g. (Sohrabi et al., 2024).

*Penalty methods* such as IPO (Liu et al., 2020) and P3O (Zhang et al., 2022) propose weighted penalty-based policy optimization objectives, where the penalties are weighted against the reward objective using a weighting hyper-parameter instead of a learnable one. This simplifies the Lagrangian approach since the penalty coefficients don't have to be optimized during training, which results in improved stability. More recently, the approach to use (smoothed) log-barriers (Usmanova et al., 2024; Zhang et al., 2024a; Dey et al., 2024) became more popular, which keeps the algorithm simple due to the penalty approach, but can offer certain constraint satisfaction guarantees, see e.g. (Ni & Kamgarpour, 2024). However, working with an explicit penalty produces suboptimal policies w.r.t the original constrained MDP and thus introduces an additional error, which has to be controlled; see for example (Geist et al., 2019; Müller & Cayci, 2024) for treatments of the regularization error in the unconstrained case, and (Liu et al., 2020) for an example of an optimization gap in safe policy optimization. In contrast, combining trust region-based updates as in TRPO (Schulman et al., 2017a) with constrained optimization techniques does not change the objective function and the set of optimizers, and therefore does not introduce an additional error.

*Trust region methods* are closely related to our approach, in particular Constrained Policy Optimization (CPO; (Achiam et al., 2017)), which extends TRPO by restricting updates to the intersection of the trust region and the safe policy set, ensuring safety during training. While CPO guarantees constraint satisfaction in the infinite sample limit, in practice it tends to oscillate around the constraint boundary with high overshoot, because it relies on noisy cost advantage estimates, and because the constraint only enters the optimization problem when the iterate is close to the boundary of the safe policy set. To address constraint satisfaction, projection-based CPO (PCPO; (Yang et al., 2020)) projects updates into the safe policy space between updates, improving stability but further hindering reward maximization. Building on PCPO, (Zhang et al., 2020) and (Yang et al., 2022) also introduce projection-based approaches based on first-order updates.

**Rethinking Safe Trust Region Methods**    We adopt a trust region approach that constructs trust regions exclusively within the safe policy set, eliminating the need for projections or constrained optimization in the inner loop. Trust region methods retain TRPO's update guarantees for both reward and constraints but often underperform compared to barrier penalty methods in terms of constraint satisfaction. To address this, we replace the state-average KL-divergence with policy divergences that act as barrier functions, see Figure 1. This modification encourages updates of the resulting trust region method to move more parallel to the constraint surfaces rather than directly toward and thereby improves constraint satisfaction, simplifies optimization, and achieves competitive returns by maintaining policies within the safe set for longer, see also Figures 5 and 7 in the Appendix.

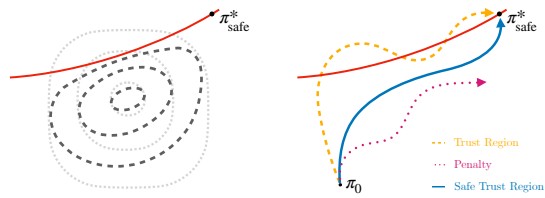

Figure 1: On the left, vanilla trust regions (dotted) and safe trust regions (dashed) are shown; on the right, a schematic visualization of common failure modes in CMDPs is shown based on optimization trajectories; here, vanilla trust regions can suffer from oscillations around the constraint, and penalty methods might introduce a bias. Lagrangian methods (not shown) can exhibit both issues.

**Contributions**    We summarize our contributions:

- In Section 3, we introduce a modified policy divergence such that every trust region consists of only safe policies. We introduce an idealized TRPO update based on the modified divergence, a tractable optimization algorithm for deep function approximation (C-TRPO), and a corresponding natural gradient method (C-NPG).

- We provide an efficient implementation of the proposed approximate C-TRPO method, see Section 3.2, which comes with a minimal overhead compared to TRPO (up to the estimation of the expected cost) and no overhead compared to CPO. We demonstrate experimentally that C-TRPO yields competitive returns with smaller constraint violations compared to common safe policy optimization algorithms, see Section 5.

- In Section 4, we introduce C-TRPO's improvement guarantees and contrast to TRPO and CPO. Further, we show that C-NPG is the continuous time limit of C-TRPO and provides global convergence guarantees towards the optimal safe policy; this is in contrast to penalization or barrier methods, which introduce a bias.

## 2. Background

We consider the infinite-horizon discounted constrained Markov decision process (CMDP) and refer the reader to (Altman, 1999) for a general treatment. The CMDP is given by the tuple $(\mathcal{S}, \mathcal{A}, P, r, \mu, \gamma, \mathcal{C})$, where $\mathcal{S}$ and $\mathcal{A}$ are the finite state-space and action-space respectively and we refer to Appendix B.3 for a discussion of continuous state and action spaces. Further, $P \colon \mathcal{S} \times \mathcal{A} \to \Delta_{\mathcal{S}}$ is the transition kernel, $r \colon \mathcal{S} \times \mathcal{A} \to \mathbb{R}$ is the reward function, $\mu \in \Delta_{\mathcal{S}}$ is the initial state distribution at time $t = 0$, and $\gamma \in [0, 1)$ is the discount factor. The space $\Delta_{\mathcal{S}}$ is the set of categorical distributions over $\mathcal{S}$. Further, define the constraint set $\mathcal{C} = \{(c_i, b_i)\}_{i=1}^m$, where $c_i \colon \mathcal{S} \times \mathcal{A} \to \mathbb{R}$ are the cost functions and $b_i \in \mathbb{R}$ are the cost thresholds.

An agent interacts with the CMDP by selecting a policy $\pi \in \Pi$ from the set of all Markov policies, i.e. an element from the Cartesian product of $|\mathcal{S}|$ probability simplices on $\mathcal{A}$. Given such a policy $\pi$, the value functions $V_r^\pi, V_{c_i}^\pi \colon \mathcal{S} \to \mathbb{R}$, action-value functions $Q_r^\pi, Q_{c_i}^\pi \colon \mathcal{S} \times \mathcal{A} \to \mathbb{R}$, and advantage functions $A_r^\pi, A_c^\pi \colon \mathcal{S} \times \mathcal{A} \to \mathbb{R}$ associated with the reward $r$ and the $i$-th cost $c_i$ are defined as

$$V_f^\pi(s) \coloneqq (1 - \gamma) \, \mathbb{E}_\pi \left[ \sum_{t=0}^\infty \gamma^t f(s_t, a_t) \Big| s_0 = s \right],$$

where the function $f$ is either $r$ or $c_i$, and the expectations are taken over trajectories of the Markov process, meaning with respect to the initial distribution $s_0 \sim \mu$, the policy $a_t \sim \pi(\cdot|s_t)$ and the state transition $s_{t+1} \sim P(\cdot|s_t, a_t)$.

Analogously, we set

$$Q_f^\pi(s, a) \coloneqq (1 - \gamma) \, \mathbb{E}_\pi \left[ \sum_{t=0}^\infty \gamma^t f(s_t, a_t) \Big| s_0 = s, a_0 = a \right]$$

and $A_f^\pi(s, a) \coloneqq Q_f^\pi(s, a) - V_f^\pi(s)$. Constrained Markov decision processes address the optimization problem

$$\text{maximize}_{\pi \in \Pi} \; V_r^\pi(\mu) \quad \text{subject to} \quad V_{c_i}^\pi(\mu) \le b_i \quad (1)$$

for all $i = 1, \dots, m$, where $V_f^\pi(\mu)$ are the expected values under the initial state distribution $V_f^\pi(\mu) \coloneqq \mathbb{E}_{s \sim \mu}[V_f^\pi(s)]$. We will also write $V_f^\pi = V_f^\pi(\mu)$, and omit the explicit dependence on $\mu$ for convenience, and we write $V_f(\pi)$ when we want to emphasize its dependence on $\pi$. We denote the set of safe policies by $\Pi_{\text{safe}} = \bigcap_{i=1}^m \{\pi : V_{c_i}(\pi) \le b_i\}$ and always assume that it is nontrivial.

**Cost Regret**   Depending on the task at hand, it is mandatory to solve the constrained optimization problem Equation 1 in a safe way, meaning with a method that respects the constraints during optimization. This motivates the use of the *cost regret*

$$\text{COSTREG}_+(\boldsymbol{\pi}, K, \mathcal{C}) \coloneqq \sum_{i=0}^m \sum_{k=0}^{K-1} \left[ V_{c_i}^{\pi_k} - b_i \right]_+, \quad (2)$$

where $[x]_+ = \max\{0, x\}$, $\boldsymbol{\pi} = (\pi_0, \pi_1, \dots \pi_K)$, and $K$ is the number of training iterations. The cost regret represents the cumulative sum of the expected constraint violations throughout training. A similar metric has been used in related online optimization settings, see (Efroni et al., 2020; Müller et al., 2024). It is our goal to design a method that produces solutions of (1) of similar quality compared to existing method, while achieving minimal cost regret.

**The Dual Linear Program for CMDPs**   Any stationary policy $\pi$ induces a discounted state-action (occupancy) measure $d_\pi \in \Delta_{\mathcal{S} \times \mathcal{A}}$, indicating the relative frequencies of visiting a state-action pair, discounted by how far the event lies in the future. This probability measure is defined as

$$d_\pi(s, a) \coloneqq (1 - \gamma) \sum_{t=0}^\infty \gamma^t \mathbb{P}_\pi(s_t = s) \pi(a|s), \quad (3)$$

where $\mathbb{P}_\pi(s_t = s)$ is the probability of observing the environment in state $s$ at time $t$ given the agent follows policy $\pi$. For finite MDPs, it is well-known that maximizing the expected discounted return can be expressed as the linear program

$$\text{maximize}_d \; r^\top d \quad \text{subject to } d \in \mathscr{D},$$

where $\mathscr{D}$ is the set of feasible state-action measures, which form a polytope (Kallenberg, 1994). Analogously to an MDP, the discounted cost CMDP can be expressed as the linear program

$$\text{maximize}_d \; r^\top d \quad \text{subject to } d \in \mathscr{D}_{\text{safe}}, \quad (4)$$

where $\mathscr{D}_{\text{safe}} = \bigcap_{i=1}^m \{d : c_i^\top d \le b_i\} \cap \mathscr{D}$ is the safe occupancy set, see Figure 4 in Appendix A.

**Information Geometry of Policy Optimization**   Among the most successful policy optimization schemes are natural policy gradient (NPG) methods or variants thereof, such as trust-region and proximal policy optimization (TRPO and PPO, respectively). These methods assume a convex geometry and corresponding Bregman divergences in the state-action polytope, see (Neu et al., 2017; Müller & Montúfar, 2023) for more detailed discussions. A general trust region update is defined as

$$\pi_{k+1} \in \arg\max_{\pi \in \Pi} \mathbb{A}_r^{\pi_k}(\pi) \quad \text{sbj. to } D_\Phi(d_{\pi_k} || d_\pi) \le \delta, \quad (5)$$

where $D_\Phi \colon \mathscr{D} \times \mathscr{D} \to \mathbb{R}$ is the Bregman divergence induced by a convex $\Phi \colon \text{int}(\mathscr{D}) \to \mathbb{R}$, and

$$\mathbb{A}_r^{\pi_k}(\pi) = \mathbb{E}_{s, a \sim d_{\pi_k}} \left[ \frac{\pi(a|s)}{\pi_k(a|s)} A_r^{\pi_k}(s, a) \right], \quad (6)$$

is called the *policy advantage* or *surrogate advantage*. We can interpret $\mathbb{A}$ as a surrogate optimization objective for the

expected return. In particular, for a parameterized policy $\pi_\theta$, it holds that $\nabla_\theta \mathbb{A}_{r,\pi_{\theta_k}}(\pi_\theta)|_{\theta=\theta_k} = \nabla_\theta V_r(\theta_k)$, see (Kakade & Langford, 2002; Schulman et al., 2017a).

TRPO and the original NPG assume the same policy geometry (Kakade, 2001; Schulman et al., 2017a), since they employ an identical Bregman divergence

$$D_K(d_{\pi_1}||d_{\pi_2}) := \sum_s d_{\pi_1}(s) D_{KL}(\pi_1(\cdot|s)||\pi_2(\cdot|s)).$$

We refer to Appendix A for details on Bregman divergences. We call $D_K$ the *Kakade divergence* and informally write $D_K(\pi_1, \pi_2) := D_K(d_{\pi_1}, d_{\pi_2})$. This divergence can be shown to be the Bregman divergence induced by the negative conditional entropy

$$\Phi_K(d_\pi) := \sum_{s,a} d_\pi(s,a) \log \pi(a|s), \qquad (7)$$

see (Neu et al., 2017). It is well known that with a parameterized policy $\pi_\theta$, a linear approximation of $\mathbb{A}$ and a quadratic approximation of the Bregman divergence $D_K$ at $\theta_k$, one obtains the *natural policy gradient* step given by

$$\theta_{k+1} = \theta_k + \epsilon_k G_K(\theta_k)^+ \nabla_\theta V_r(\pi_{\theta_k}), \qquad (8)$$

where $G_K(\theta)^+$ denotes a pseudo-inverse of the generalized Fisher-information matrix of the policy with entries given by $G_K(\theta)_{ij} = \partial_{\theta_i} d_\theta \nabla^2 \Phi_K(d_\theta) \partial_{\theta_j} d_\theta$, see (Schulman et al., 2017a; Müller & Montúfar, 2023) and Appendix A for more detailed discussions.

## 3. A Safe Geometry for Constrained MDPs

To prevent the policy iterates from violating the constraints during optimization, we construct policy divergences for which the trust regions are contained in the safe policy set.

### 3.1. Safe Trust Regions

A Bregman divergence is induced by a mirror function that dictates the behavior of the divergence, see Appendix A. Take for example the mirror function for TRPO and NPG in Equation (7). The divergence is defined when both policies are in the interior of $\mathscr{D}$, and as either one of the policies approaches the boundary of the state-action polytope, the divergence approaches infinity. Hence, TRPO and NPG don't allow their policy iterates to become entirely deterministic during optimization.

Since the behavior of a Bregman divergences is dictated by the shape of its mirror function, we first construct a family of *safe mirror functions*, that induce policy divergences that are finite only in the safe occupancy set $\mathscr{D}_{safe}$ instead of the entire state-action polytope $\mathscr{D}$. Safe policy divergences, in turn, let us derive safe trust region and natural policy gradient methods.

To this end, we consider mirror functions of the form

$$\Phi_C(d) := \Phi_K(d) + \sum_{i=1}^m \beta_i \phi(b_i - c_i^\top d), \qquad (9)$$

where $\Phi_K$ is the conditional entropy defined in Equation (7), and $\phi: \mathbb{R}_{>0} \to \mathbb{R}$ is a convex function with $\phi'(x) \to +\infty$ for $x \searrow 0$. This ensures that $\Phi_C: \text{int}(\mathscr{D}_{safe}) \to \mathbb{R}$ is strictly convex and has infinite curvature at the cost surface $b_i - c_i^\top d = 0$, which means $\|\nabla \Phi_C(d_k)\| \to +\infty$, when $b_i - c_i^\top d_k \searrow 0$. Possible candidates for $\phi$ are $\phi(x) = -\log(x)$ and $\phi(x) = x \log(x)$ corresponding to a logarithmic barrier and entropy, respectively.

The mirror function $\Phi_C$ induces the *Constrained KL-Divergence* given by

$$D_C(d_1||d_2) = D_K(d_1||d_2) + \sum_{i=1}^m \beta_i D_{\phi_i}(d_1||d_2), \quad (10)$$

where

$$\begin{aligned} D_{\phi_i}(d_1||d_2) = &\phi(b_i - V_{c_i}(\pi_1)) - \phi(b_i - V_{c_i}(\pi_2)) \\ &+ \phi'(b_i - V_{c_i}(\pi_2))(V_{c_i}(\pi_1) - V_{c_i}(\pi_2)). \end{aligned}$$
(11)

The corresponding trust-region scheme is

$$\pi_{k+1} \in \arg\max_{\pi \in \Pi} \mathbb{A}_r^{\pi_k}(\pi) \quad \text{sbj. to } D_C(d_{\pi_k}||d_\pi) \leq \delta,$$
(12)

where $\mathbb{A}_r$ is defined in Equation (6). Note the constraint is only satisfied if $d_1, d_2 \in \text{int}(\mathscr{D}_{safe})$ and the divergence approaches $+\infty$ as $d_2$ approaches the boundary of the safe set. Thus, the trust region $\{d \in \mathscr{D} : D_C(d_k||d) \leq \delta\}$ is contained in the set of safe occupancy measures for any finite $\delta$. Analogously to the case of unconstrained TRPO the corresponding natural policy gradient scheme is

$$\theta_{k+1} = \theta_k + \epsilon_k G_C(\theta_k)^+ \nabla V_r(\theta_k), \qquad (13)$$

where $G_C(\theta)^+$ denotes an arbitrary pseudo-inverse and

$$G_C(\theta)_{ij} = \partial_{\theta_i} d_\theta^\top \nabla^2 \Phi_C(d_\theta) \partial_{\theta_j} d_\theta.$$

### 3.2. Constrained Trust Region Policy Optimization

If we could solve the optimization problem in Equation (12) exactly, we would obtain a provably safe trust region policy optimization method with zero constraint violations, as long as we start with a safe policy. However, the exact trust region update Equation (12) cannot be computed. Firstly, the divergence depends on expected cost values, which we can only estimate. The resulting estimation errors of the divergence might cause the policy iterates to leave the safe set, in which case the divergence becomes ill-defined. Further, the

divergence also depends on the expected cost value of the proposal policy, which is not available during the updates. To address these issues, we propose an update based on a *surrogate divergence*, similar to how surrogate objectives are used in policy optimization. We propose the following update, which we call *Constrained TRPO* (C-TRPO).

$$\pi_{k+1} = \arg\max_{\pi \in \Pi} \mathbb{A}_r^{\pi_k}(\pi) \quad \text{sbj. to } \bar{D}_C(\pi||\pi_k) \le \delta. \quad (14)$$

Here, $\bar{D}_C$ is a surrogate for $D_C$, which we define in Equation 16 and Equation 17. Algorithm 1 shows the implementation of C-TRPO, which performs a constrained trust region update if the current policy is safe or a recovery step that minimizes the cost if the policy is unsafe. For the trust region update, we follow a similar implementation to the original TRPO, estimating the divergence, using a linear approximation of the surrogate objective, and a quadratic approximation of the trust region.

**Surrogate Divergence**  For the sake of clarity, we first focus on the case with a single constraint, but the results are easily extended to multiple constraints by summation of the individual constraint terms, as discussed in the respective paragraph below. In practice, the exact constrained KL-Divergence $D_C$ cannot be evaluated, because it depends on the cost-return of the optimized policy $V_c(\pi)$. However, we can approximate it locally around the policy of the $k$-th iteration, $\pi_k$, using a surrogate divergence. This surrogate can be expressed as a function of the policy cost advantage

$$\mathbb{A}_c^{\pi_k}(\pi) = \mathbb{E}_{d_{\pi_k}}\left[\frac{\pi(a|s)}{\pi_k(a|s)} A_c^{\pi_k}(s,a)\right], \quad (15)$$

which approximates $V_c(\pi) - V_c^{\pi_k}$ up to first order in the policy parameters (Kakade & Langford, 2002; Schulman et al., 2017a; Achiam et al., 2017). Assume $\pi_k \in \Pi_{\text{safe}}$ and define the *constraint margin* $\delta_b = b - V_c^{\pi_k}$, which is positive if $\pi_k \in \Pi_{\text{SAFE}}$. Further, define the surrogate divergence $\bar{D}_C(\pi||\pi_k) = \bar{D}_{\text{KL}}(\pi||\pi_k) + \beta\bar{D}_\phi(\pi||\pi_k)$, where

$$\bar{D}_{\text{KL}}(\pi||\pi_k) = \sum_{s \in \mathcal{S}} d_{\pi_k}(s) D_{\text{KL}}(\pi||\pi_k) \quad (16)$$

and

$$\bar{D}_\phi(\pi_\theta||\pi_{\theta_k}) = \begin{cases} \Psi(\mathbb{A}_c^{\pi_k}), & \text{if } \delta_b - \mathbb{A}_c^{\pi_k} \in \text{dom}(\phi) \\ +\infty & \text{otherwise} \end{cases} \quad (17)$$

where

$$\Psi(\mathbb{A}_c^{\pi_k}) = \phi(\delta_b - \mathbb{A}_c^{\pi_k}(\pi)) - \phi(\delta_b) + \phi'(\delta_b)\mathbb{A}_c^{\pi_k}(\pi). \quad (18)$$

The surrogate $\bar{D}_\phi$ is closely related to the Bregman divergence $D_\phi$. They are equivalent up to the substitution $V_c(\pi) - V_c(\pi_k) \rightarrow \mathbb{A}_c^{\pi_k}(\pi)$, see Appendix B.1. The surrogate can be estimated from samples of the CMDP, where

in the practical implementation, $\delta_b$ and the policy cost advantage are estimated from trajectory samples using GAE-$\lambda$ (Schulman et al., 2018). The consequences of the substitution in the surrogate will be discussed in Section 4.

**Comparison with CPO**  This approach is similar to the update in CPO (Achiam et al., 2017), but incorporates the constraint into the design of the trust region, with an influence controlled by the parameter $\beta$. This yields more conservative updates within the safe set without introducing bias in the optimal solution. Additionally, it simplifies the inner-loop constrained optimization: C-TRPO approximates a single quadratic constraint, rather than solving for the intersection of a quadratic and a linear constraint as in CPO, see also Appendix C.4.

**Multiple Constraints**  C-TRPO naturally extents to multiple constraints, by introducing the divergence $\bar{D}_C^{\text{mult}}(\pi||\pi_k) = \bar{D}_{\text{KL}}(\pi||\pi_k) + \sum_i \beta_i \bar{D}_{\phi_i}(\pi||\pi_k)$, where each $\bar{D}_{\phi_i}$ is defined according to Eq. 17 but with the respective $c_i$. In section 3, we discuss that this divergence approximates a natural policy gradient (C-NPG) on the safe state-action occupancy set, where Theorem 4.5 implies that the optimal feasible solution $\pi_{\text{safe}}^\star$ satisfies as few constraints with equality as required to be optimal.

**Recovery with Hysteresis**  The iterate may still leave the safe policy set $\Pi_{\text{safe}}$, either due to approximation errors of the divergence, or because we started outside the safe set. In this case, we perform a recovery step, where we only minimize the cost with TRPO as by (Achiam et al., 2017). In tasks where the policy starts in the unsafe set, C-TRPO can get stuck at the constraint surface. This is easily mitigated by including a hysteresis condition for returning to the safe set. If $\pi_{k-1}$ is the previous policy, then $\pi_k \in \Pi_{\text{safe}}^{\text{H}}$ with $\Pi_{\text{safe}}^{\text{H}} = \{\pi_\theta \in \Pi_\theta \text{ and } V_c(\pi_\theta) \le b_{\text{H}}\}$ where $b_{\text{H}} = b$ if $\pi_{k-1} \in \Pi_{\text{safe}}^{\text{H}}$ and a user-specified fraction of $b$ otherwise.

**Computational Complexity**  The C-TRPO implementation adds no computational overhead compared to CPO, since $\bar{D}_\phi$ is a function of the cost advantage estimate, and is added to the divergence of TRPO. Compared to TRPO, the cost value function must be approximated.

### 3.3. Constrained Natural Policy Gradient

Practically, the C-TRPO optimization problem in Equation (14) is solved like traditional TRPO: the objective is approximated linearly, and the constraint is approximated quadratically in the policy parameters using automatic differentiation and the conjugate gradient method. This leads

**Algorithm 1** Constrained TRPO (C-TRPO); differences from TRPO **in blue**

1: **Input:** Initial policy $\pi_0 \in \Pi_\theta$, safety parameter $\beta > 0$, recovery parameter $0 < b_H \le b$
2: **for** $k = 0, 1, 2, \ldots$ **do**
3:     Sample a set of trajectories following $\pi_k = \pi_{\theta_k}$
4:     **if** $\pi_k \in \Pi_{\text{safe}}^H$ **then**
5:         $A \leftarrow A_r$; $D \leftarrow \bar{D}_C = \bar{D}_{\text{KL}} + \beta \bar{D}_\phi$ {Constrained trust region update}
6:     **else**
7:         $A \leftarrow -A_c$; $D \leftarrow \bar{D}_{\text{KL}}$ {Recovery}
8:     **end if**
9:     Compute $\pi_{k+1}$ using TRPO with $A$ as advantage estimate and with $D$ as policy divergence.
10: **end for**

to the policy parameter update

$$\theta_{k+1} = \theta_k + \alpha^i \sqrt{\frac{2\delta}{g_k^\top H_k^{-1} g_k}} \cdot H_k^{-1} g_k, \qquad (19)$$

where

$$g_k = \nabla_\theta \mathbb{A}_c^{\theta_k}(\pi_\theta)|_{\theta = \theta_k} \qquad (20)$$

and

$$H_k = \bar{H}_C(\theta_k) = \nabla_\theta^2 \bar{D}_C(\pi_\theta || \pi_{\theta_k})|_{\theta = \theta_k} \qquad (21)$$

are finite sample estimates, and $H^{-1} g$ is approximated using conjugate gradients. The $\alpha^i \in [0, 1]$ are the coefficients for backtracking line search, which ensures $\bar{D}_C(\pi_\theta || \pi_{\theta_k}) \le \delta$.

We show in Appendix B.2.3 that the Hessian

$$\bar{H}_C(\theta_k) = G_K(\theta_k) + \beta \phi''(b - V_c^{\hat\theta}(\theta)) \nabla_\theta V_c^{\hat\theta}(\theta) \nabla_\theta V_c^{\hat\theta}(\theta)^\top,$$

is equivalent to the Gramian $G_C(\theta_k)$ of the natural gradient update in Equation (13). We call the resulting policy gradient

$$\theta_{k+1} = \theta_k + \epsilon_k \bar{H}_C(\theta_k)^+ \nabla V_r(\theta_k), \qquad (22)$$

the *Constrained NPG* (C-NPG). In particular, this shows that the C-TRPO update can be interpreted as a natural policy gradient step with an adaptive step size, see Appendix A. We emphasize that the idealized safe trust region update in Equation (12) and the C-TRPO update of Equation (14) agree up to second order in the policy parameters. This justifies the surrogate divergence in C-TRPO and motivates the discussion of the C-NPG flow in Section 4.2. We show in Theorem 4.4 that $\text{int}(\mathscr{D}_{\text{safe}})$ is invariant under the dynamics of the C-NPG. This implies that if the trust region radius $\delta$ is small, and the advantage estimation is accurate enough, the iterates under C-TRPO never leave the safe set.

## 4. Analysis

Here, we provide a theoretical analysis of the updates of C-TRPO and study the convergence properties of the time-continuous version of C-NPG. All proofs are deferred to the Appendix C.

### 4.1. Properties of the C-TRPO Update

The practical C-TRPO algorithm is implemented using the surrogate divergence introduced in Equation (14), which is identical to the theoretical divergence $D_C$ introduced in Equation (12) up to a mismatch between the policy advantage and the performance difference. The motivation for substituting the policy cost advantage for the performance difference is their equivalence up to first order and that we can estimate the advantage from samples of $d_\pi$. Similar to CPO, we can guarantee an almost improvement of the return (Achiam et al., 2017), despite the new divergence.

**Proposition 4.1** (C-TRPO reward update). *Set* $\epsilon_r = \max_s |\mathbb{E}_{a \sim \pi_{k+1}} A_r^{\pi_k}(s, a)|$. *The expected reward of a policy updated with C-TRPO is bounded from below by*

$$V_r(\pi_{k+1}) \ge V_r(\pi_k) - \frac{\sqrt{2\delta} \gamma \epsilon_r}{1 - \gamma}. \qquad (23)$$

Constraint violation, however, behaves slightly differently for the two algorithms. To see this, we establish a more concrete relation between C-TRPO and CPO. As $\beta \searrow 0$, the solution to Equation (14) approaches the constraint surface in the worst case, and we recover CPO, see Figure 5.

**Proposition 4.2.** *The approximate C-TRPO update approaches the CPO update in the limit as* $\beta \searrow 0$.

Intuitively, solving the C-TRPO problem with successively smaller values of $\beta$, would be similar to CPO with the interior point method using $\bar{D}_\phi(\cdot || \pi_k)$ as the barrier function. However, C-TRPO is more conservative than CPO for any $\beta > 0$ and as $\beta \to +\infty$ the update is maximally constrained in the cost-increasing direction.

**Proposition 4.3** (C-TRPO worst-case constraint violation). *Let* $\bar{D}_C(\pi_{k+1} || \pi_k) \le \delta$ *with* $\delta > 0$ *and set* $\epsilon_c = \max_s |\mathbb{E}_{a \sim \pi_{k+1}} A_c^{\pi_k}(s, a)|$. *It holds that*

$$V_c(\pi_{k+1}) \le V_c(\pi_k) + \mathbb{A}_c^{\pi_k}(\pi_{k+1}) + \frac{\sqrt{2\delta(\beta)} \gamma \epsilon_c}{1 - \gamma}, \quad (24)$$

*where* $\delta(\beta) = \delta - \beta \bar{D}_\phi(\pi_{k+1}, \pi_k) \le \delta$ *is decreasing in* $\beta > 0$, $\lim_{\beta \to 0} \delta(\beta) = \delta$, *and* $\delta(\beta) \to 0$ *for* $\beta \to \delta \bar{D}_C(\pi_{k+1} || \pi_k) / \bar{D}_\phi(\pi_{k+1}, \pi_k)$.

This result is analogous to the worst-case constraint violation for CPO (Achiam et al., 2017, Proposition 2), where the term $\delta(\beta)$ is replaced by $\delta$. As $\delta(\beta) \le \delta$ for all $\beta > 0$, the bound for C-TRPO is higher than the corresponding

guarantee for CPO. For $\beta \to 0$, the bound converges to the CPO bound, where for $\beta \to +\infty$, the bound becomes $V_c(\pi_{k+1}) \leq V_c(\pi_k)$, see Appendix C.1.

## 4.2. Invariance and Convergence of Constrained Natural Policy Gradients

It is well known that TRPO is equivalent to a natural policy gradient method with an adaptive step size, see also Appendix A. We study the time-continuous limit of C-TRPO and guarantee safety during training and global convergence. In the context of constrained TRPO in Equation (12), we study the natural policy gradient flow

$$\partial_t \theta_t = G_{\mathrm{C}}(\theta_t)^+ \nabla V_r(\theta_t), \qquad (25)$$

where $G_{\mathrm{C}}(\theta)^+$ denotes a pseudo-inverse of $G_{\mathrm{C}}(\theta)_{ij} = \partial_{\theta_i} d_\theta^\top \nabla^2 \Phi_{\mathrm{C}}(d_\theta) \partial_{\theta_j} d_\theta$ and $\theta \mapsto \pi_\theta$ is a differentiable policy parametrization. Moreover, we assume that $\theta \mapsto \pi_\theta$ is regular, that is it is surjective and the Jacobian is of maximal rank everywhere. This assumption implies overparametrization but is satisfied for common models like tabular softmax, tabular escort, or expressive log-linear policy parameterizations (Agarwal et al., 2021a; Mei et al., 2020a; Müller & Montúfar, 2023).

We denote the set of safe parameters by $\Theta_{\mathrm{safe}} := \{\theta \in \mathbb{R}^p : \pi_\theta \in \Pi_{\mathrm{safe}}\}$, which is non-convex in general and say that $\Theta_{\mathrm{safe}}$ is *invariant* under Equation (25) if $\theta_0 \in \Theta_{\mathrm{safe}}$ implies $\theta_t \in \Theta_{\mathrm{safe}}$ for all $t$. Invariance is associated with safe control during optimization and is typically achieved via control barrier function methods (Ames et al., 2017; Cheng et al., 2019). We study the evolution of the state-action distributions $d_t = d^{\pi_{\theta_t}}$ as this allows us to employ the linear programming formulation of CMDPs and we obtain the following convergence guarantees.

**Theorem 4.4** (Safety during training). *Assume that $\phi \colon \mathbb{R}_{>0} \to \mathbb{R}$ satisfies $\phi'(x) \to +\infty$ for $x \searrow 0$ and consider a regular policy parameterization. Then the set $\Theta_{\mathrm{safe}}$ is invariant under Equation (25).*

A visualization of policies obtained by C-NPG for different safe initializations and varying choices of $\beta$ is shown in Figure 2 for a toy MDP. We see that for even small choices of $\beta$ the trajectories don't cross the constraint surface and the updates become more conservative for larger choices of $\beta$.

**Theorem 4.5.** *Assume that $\phi'(x) \to +\infty$ for $x \searrow 0$, set $V_{r,\mathrm{C}}^\star := \max_{\pi \in \Pi_{\mathrm{safe}}} V_r(\pi)$ and denote the set of optimal constrained policies by $\Pi_{\mathrm{safe}}^\star = \{\pi \in \Pi_{\mathrm{safe}} : V_r(\pi) = V_{r,\mathrm{C}}^\star\}$, consider a regular policy parametrization and let $(\theta_t)_{t \geq 0}$ solve Equation (25). It holds that $V_r(\pi_{\theta_t}) \to V_{r,\mathrm{C}}^\star$ and*

$$\lim_{t \to +\infty} \pi_t = \pi_{\mathrm{safe}}^\star = \arg\min\{D_{\mathrm{C}}(\pi^\star, \pi_0) : \pi^\star \in \Pi_{\mathrm{safe}}^\star\}. \qquad (26)$$

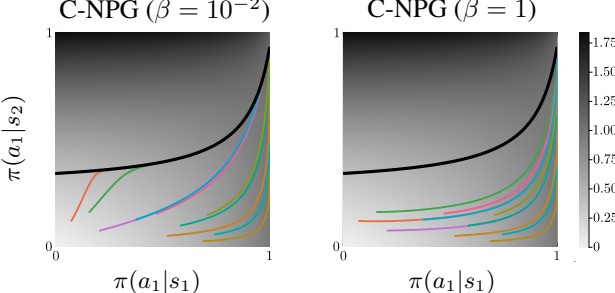

Figure 2: Shown is the policy set $\Pi \cong [0,1]^2$ for an MDP with two states and two actions with a heatmap of the expected reward $V_r$; the constraint surface is shown in black with the safe policies below; optimization trajectories are shown for 10 safe initialization and for $\beta = 10^{-2}, 1$.

In case of multiple optimal policies, Equation (26) identifies the optimal policy of the CMDP that the natural policy gradient method converges to as the projection of the initial policy $\pi_0$ to the set of optimal safe policies $\Pi_{\mathrm{safe}}^\star$ with respect to the constrained divergence $D_{\mathrm{C}}$. In particular, this implies that the limiting policy $\pi_{\mathrm{safe}}^\star$ satisfies as few constraints with equality as required to be optimal. To see this, note that $\Pi_{\mathrm{safe}}^\star$ forms a face of $\mathscr{D}_{\mathrm{safe}}$ and that Bregman projections lie at the interior of faces (Müller et al., 2024, Lemma A.2) and hence satisfy as few linear constraints as required.

## 5. Computational Experiments

**Setup and main results** We benchmark C-TRPO against 9 common safe policy optimization algorithms (CPO (Achiam et al., 2017), PCPO (Yang et al., 2020), CPPO-PID (Stooke et al., 2020), PPO-Lag and TRPO-Lag (Achiam et al., 2017; Ray et al., 2019), FOCOPS (Zhang et al., 2020), CUP (Yang et al., 2022), IPO (Liu et al., 2020) and P3O (Zhang et al., 2022)) on 8 tasks (4 Navigation and 4 Locomotion) from the Safety Gymnasium (Ji et al., 2023) benchmark.[1] The locomotion tasks reward distance traveled, while penalizing high velocities, and the navigation tasks reward goal reaching and penalize certain unsafe states. For the C-TRPO implementation we fix the convex generator $\phi(x) = x \log(x)$, motivated by its superior performance in our experiments, see Appendix B.2.1, and $b_{\mathrm{H}} = 0.8b$ and $\beta = 1$ across all experiments. Each algorithm is evaluated by training for 10 million environment steps with 5 seeds each, and the cost regret is monitored throughout training for every run. To get a better sense of the safety of the algorithms during training, we take an online learning perspective and include as a metric the cost regret introduced in Equation 2 (Efroni et al., 2020; Müller et al., 2024) For

[1] Code: https://github.com/milosen/ctrpo

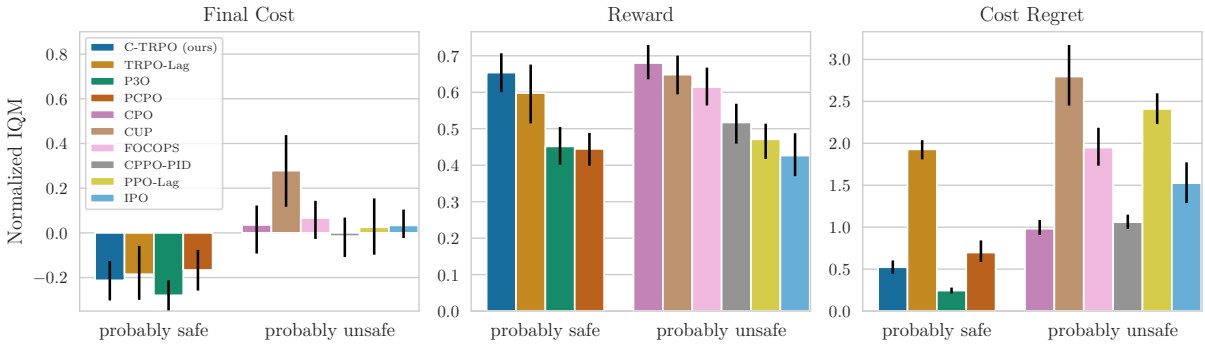

Figure 3: Comparison of safe policy optimization algorithms based on the 25% Inter Quartile Mean (IQM) across 5 seeds and 8 tasks. From left to right, the following metrics are shown measured at 10 million training steps: the *final cost*, i.e. the mean return of the cost at the last iterate (threshold-normalized and centered at zero), the mean return of the *reward* (normalized with the performance of unconstrained PPO), and the mean *cost regret* (normalized by CPO's cost regret). The algorithms are sorted into *probably safe* and *probably unsafe*, based on their final constraint violation (negative is probably safe), and by expected reward within each group. Note that cost regret is different from the final cost, since it sums up all constraint violations throughout training.

completeness, we also report environment-wise sample efficiency curves and the results of Figure 3 in a tabular format in Appendix D.4.

**Discussion** In Figure 3 the interquartile mean (IQM) of normalized expected reward, cost, and cost regret, including their stratified bootstrap confidence intervals (Agarwal et al., 2021b) is shown. It can be observed that C-TRPO is competitive with the leading algorithms of the benchmark in terms of expected return, while being safe on the last iterate as opposed to CPO and CUP, see Figure 3. Furthermore, it achieves notably lower cost regret throughout training than the high-return algorithms. TRPO-Lag., which is also safe at convergence, has notably higher cost regret than the other safe methods, meaning it oscillates more around the threshold during training, see also Figures 13 and 12 in the appendix. In general, methods that, in practice, rely on Lagrangian-inspired optimization routines (TRPO-Lag., FOCOPS, and CUP) perform well in terms of reward, but poorly in terms of cost regret. C-TRPO's regret performance is comparable to the more conservative PCPO algorithm, but is not as low as that of P3O. The low cost regret achieved by P3O comes at the price of expected reward, which is due to it's wide margin to the threshold at the last iterate.

Our experiments reveal that C-TRPO's performance is closely tied to the accuracy of divergence estimation, which hinges on the precise estimation of the cost advantage and value functions. C-TRPO's behavior w.r.t noisy cost function estimates is analyzed in Appendix D.3. The safety parameter $\beta$ modulates the stringency with which C-TRPO satisfies the constraint, and can do so without limiting the expected return on most environments at least for $\beta \leq 1$, see

Figure 8 in the appendix. For higher values, the expected return starts to degrade, partly due to $\bar{D}_\phi$ being relatively noisy compared to $\bar{D}_{\mathrm{KL}}$ and thus we recommend the choice $\beta = 1$.

Further, we observe that in most environments constraint violations seem to reduce as the algorithm converges, meaning that the regret flattens over time. This behavior suggests that the divergence estimation becomes increasingly accurate over time, potentially allowing C-TRPO to achieve sublinear regret. However, we leave regret analysis of the finite sample regime for future research.

We attribute the improved constraint satisfaction compared to CPO to a slowdown and reduction in the frequency of oscillations around the cost threshold, which mitigates overshoot behaviors that could otherwise violate constraints. The modified gradient preconditioner appears to deflect the parameter trajectory away from the constraint, see Figure 2. This effect may also be partially attributed to the hysteresis-based recovery mechanism, which helps smooth updates by leading the iterate away from the boundary of the safe set. Employing a hysteresis fraction $0 < b_{\mathrm{H}} < b$ might also be beneficial because C-TRPO's divergence estimates tend to be more reliable for strictly safe policies. The effect of the choice of $b_{\mathrm{H}}$ is shown in Figure 9 in the appendix. Finally, we present ablations in Appendix D.2, which support our claims that both components—the modified trust region and hysteresis—are effective in reducing safety violations.

## 6. Conclusion and outlook

We introduced C-TRPO and C-NPG, two novel methods for solving CMDPs. C-TRPO extends Trust Region Policy

Optimization (TRPO) by embedding constraint handling into the policy space geometry, while C-NPG provides a provably safe natural policy gradient method for CMDPs. Our experiments showed that C-TRPO reduces constraint violations while maintaining competitive returns compared to state-of-the-art constrained RL algorithms. Despite these advances, challenges remain. Estimating the proposed divergence is difficult, and we did not analyze its finite-sample properties. Additionally, CMDPs constrain average cost return, making trajectory-wise or state-wise safety constraints harder to model. Future work includes integrating C-TRPO with model-based methods (As et al., 2025), leveraging mirror descent (Tomar et al., 2022), and considering alternative formalisms for risk-sensitive RL like distributional RL (Dabney et al., 2018) or Sauté RL (Sootla et al., 2022).

Overall, the proposed algorithms, C-TRPO and C-NPG, present a step forward in general-purpose CMDP algorithms and move us closer to deploying RL in high-stakes, real-world applications.

## Impact Statement

This paper presents work whose goal is to advance the field of constrained Markov decision processes and safe reinforcement learning. There are many potential societal consequences of our work, none of which we feel must be specifically highlighted here.

## Acknowledgements

N. M. and N.S. are supported by BMBF (Federal Ministry of Education and Research) through ACONITE (01IS22065) and the Center for Scalable Data Analytics and Artificial Intelligence (ScaDS.AI.) Leipzig and by the European Union and the Free State of Saxony through BIOWIN. N.M. is also supported by the Max Planck IMPRS CoNI Doctoral Program.

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

# A. Extended Background

We consider the infinite-horizon discounted Markov decision process (MDP), given by the tuple $(\mathcal{S}, \mathcal{A}, P, r, \mu, \gamma)$. Here, $\mathcal{S}$ and $\mathcal{A}$ are the finite state-space and action-space respectively. Here, we make the restriction to finite MDPs as this simplifies the presentation. For a discussion of continuous state and action spaces, we refer to Appendix B.3. Further, $P \colon \mathcal{S} \times \mathcal{A} \to \Delta_{\mathcal{S}}$ is the transition kernel, $r \colon \mathcal{S} \times \mathcal{A} \to \mathbb{R}$ is the reward function, $\mu \in \Delta_{\mathcal{S}}$ is the initial state distribution at time $t = 0$, and $\gamma \in [0, 1)$ is the discount factor. The space $\Delta_{\mathcal{S}}$ is the set of categorical distributions over $\mathcal{S}$.

The Reinforcement Learning (RL) protocol is usually described as follows: At time $t = 0$, an initial state $s_0$ is drawn from $\mu$. At each integer time-step $t$, the agent chooses an action according to it's (stochastic) behavior policy $a_t \sim \pi(\cdot|s_t)$. A reward $r_t = r(s_t, a_t)$ is given to the agent, and a new state $s_{t+1} \sim P(\cdot|s_t, a_t)$ is sampled from the environment. Given a policy $\pi$, the value function $V_r^\pi \colon \mathcal{S} \to \mathbb{R}$, action-value function $Q_r^\pi \colon \mathcal{S} \times \mathcal{A} \to \mathbb{R}$, and advantage function $A_r^\pi \colon \mathcal{S} \times \mathcal{A} \to \mathbb{R}$ associated with the reward $r$ are defined as

$$V_r^\pi(s) := (1 - \gamma) \, \mathbb{E}_\pi \left[ \sum_{t=0}^{\infty} \gamma^t r(s_t, a_t) \Big| s_0 = s \right],$$

$$Q_r^\pi(s, a) := (1 - \gamma) \, \mathbb{E}_\pi \left[ \sum_{t=0}^{\infty} \gamma^t r(s_t, a_t) \Big| s_0 = s, a_0 = a \right] \text{ and } A_r^\pi(s, a) := Q_r^\pi(s, a) - V_r^\pi(s).$$

where and the expectations are taken over trajectories of the Markov process resulting from starting at $s$ and following policy $\pi$. The goal is to

$$\text{maximize}_{\pi \in \Pi} \; V_r^\pi(\mu) \tag{27}$$

where $V_r^\pi(\mu)$ is the expected value under the initial state distribution $V_r^\pi(\mu) := \mathbb{E}_{s \sim \mu}[V_r^\pi(s)]$. We will also write $V_r^\pi = V_r^\pi(\mu)$, and omit the explicit dependence on $\mu$ for convenience, and we write $V_r(\pi)$ when we want to emphasize its dependence on $\pi$.

**The Dual Linear Program for MDPs**  Any stationary policy $\pi$ induces a discounted state-action (occupancy) measure $d_\pi \in \Delta_{\mathcal{S} \times \mathcal{A}}$, indicating the relative frequencies of visiting a state-action pair, discounted by how far the visitation lies in the future. It is a probability measure defined as

$$d_\pi(s, a) := (1 - \gamma) \sum_{t=0}^{\infty} \gamma^t \mathbb{P}_\pi(s_t = s) \pi(a|s), \tag{28}$$

where $\mathbb{P}_\pi(s_t = s)$ is the probability of observing the environment in state $s$ at time $t$ given the agent follows policy $\pi$. For finite MDPs, it is well-known that maximizing the expected discounted return can be expressed as the linear program

$$\max_d \; r^\top d \quad \text{subject to } d \in \mathscr{D}, \tag{29}$$

where $\mathscr{D}$ is the set of feasible state-action measures (Feinberg & Shwartz, 2012). This set is also known as the *state-action polytope*, defined by

$$\mathscr{D} = \left\{ d \in \mathbb{R}_{\geq 0}^{\mathcal{S} \times \mathcal{A}} : \ell_s(d) = 0 \text{ for all } s \in \mathcal{S} \right\},$$

where the linear constraints $\ell_s(d)$ are given by the *Bellman flow equations*

$$\ell_s(d) = d(s) - \gamma \sum_{s', a'} d(s', a') P(s|s', a') - (1 - \gamma) \mu(s),$$

where $d(s) = \sum_a d(s, a)$ denotes the state-marginal of $d$. For any state-action measure $d$ we obtain the associated policy via conditioning, meaning

$$\pi(a|s) := \frac{d(s, a)}{\sum_{a'} d(s, a')} \tag{30}$$

in case this is well-defined. This provides a one-to-one correspondence between policies and the state-action distributions under the following assumption.

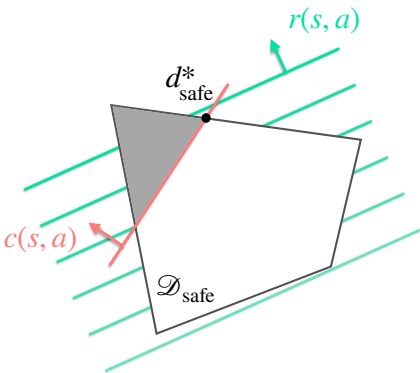

Figure 4: The dual linear program for a CMDP of two states and two actions.

**Assumption A.1** (Exploration). For any policy $\pi \in \Delta_{\mathcal{A}}^{\mathcal{S}}$ we have $d_\pi(s) > 0$ for all $s \in \mathcal{S}$.

This assumption is standard in linear programming approaches and policy gradient methods where it is necessary for global convergence (Kallenberg, 1994; Mei et al., 2020b). Note that $d \in \partial \mathcal{D}$ if and only if $d(s, a) = 0$ for some $s, a$ and hence the boundary of $\mathcal{D}$ is given by

$$\partial \mathscr{D} = \Big\{ d_\pi : \pi(a|s) = 0 \text{ for some } s \in \mathcal{S}, a \in \mathcal{A} \Big\}.$$

**Constrained Markov Decision Processes**   Where MDPs aim to maximize the return, constrained MDPs (CMDPs) aim to maximize the return subject to a number of costs not exceeding certain thresholds. For a general treatment of CMDPs, we refer the reader to (Altman, 1999). An important application of CMDPs is in safety-critical reinforcement learning where the costs incorporate safety constraints. An infinite-horizon discounted CMDP is defined by the tuple $(\mathcal{S}, \mathcal{A}, P, r, \mu, \gamma, \mathcal{C})$, consisting of the standard elements of an MDP and an additional constraint set $\mathcal{C} = \{(c_i, b_i)\}_{i=1}^m$, where $c_i \colon \mathcal{S} \times \mathcal{A} \to \mathbb{R}$ are the cost functions and $b_i \in \mathbb{R}$ are the cost thresholds.

In addition to the value functions and the advantage functions of the reward that are defined for the MDP, we define the same quantities $V_{c_i}$, $Q_{c_i}$, and $A_{c_i}$ w.r.t the $i$th cost $c_i$, simply by replacing $r$ with $c_i$. The objective is to maximize the discounted return, as before, but we restrict the space of policies to the safe policy set

$$\Pi_{\text{safe}} = \bigcap_{i=1}^m \Big\{ \pi : V_{c_i}(\pi) \le b_i \Big\}, \tag{31}$$

where

$$V_{c_i}^\pi(\mu) \coloneqq \mathbb{E}_{s \sim \mu}[V_{c_i}^\pi(s)]. \tag{32}$$

is the expected discounted cumulative cost associated with the cost function $c_i$. Like the MDP, the discounted cost CMPD can be expressed as the linear program

$$\max_d r^\top d \quad \text{sbj. to } d \in \mathscr{D}_{\text{safe}}, \tag{33}$$

where

$$\mathscr{D}_{\text{safe}} = \bigcap_{i=1}^m \Big\{ d \in \mathbb{R}^{\mathcal{S} \times \mathcal{A}} : c_i^\top d \le b_i \Big\} \cap \mathscr{D} \tag{34}$$

is the safe occupancy set, see Figure 4.

**Bregman divergences**  Here, we give a short introduction to the concept of Bregman divergences, which is required for the formulation of trust region methods. For this, we consider a convex subset of Euclidean space $C \subseteq \mathbb{R}^d$ with a non-empty interior $\mathrm{int}(C)$ and a strictly convex function $\phi \colon C \to \mathbb{R}$ which we assume to be differentiable on the interior $\mathrm{int}(C)$. Then, the *Bregman divergence* induced by $\phi$ is given by

$$D_\phi(x||y) := \phi(x) - \phi(y) - \nabla\phi(y)^\top (x - y), \tag{35}$$

which is well defined for $x \in C, y \in \mathrm{int}(C)$. Intuitively, the Bregman divergence measures the difference between $\phi$ and its linearization at $y$. The strict convexity of $\phi$ ensures that $D_\phi(x||y) \geq 0$ and $D_\phi(x||y) = 0$ if and only if $x = y$. Therefore, Bregman divergences are commonly interpreted as a generalized measure for the distance between points, however, it is important to notice that it is not generally symmetric. An important example is the Euclidean distance $D_\phi(x||y) = \|x - y\|_2^2$ which arises from the choice $\phi(x) := \|x\|_2^2$. Another important Bregman divergence is the Kullback-Leibler (KL) divergence

$$D_{\mathrm{KL}}(p||q) := \sum_{i=1}^d p_i \log \frac{p_i}{q_i} - \sum_{i=1}^d p_i + \sum_{i=1}^d q_i, \tag{36}$$

where we use the common convention $0 \log \frac{0}{0} := 0$. Then, the KL divergence is defined for $p \in \mathbb{R}_{\geq 0}^d$ and $q \in \mathbb{R}_{\geq 0}^d$ which is absolutely continuous with respect to $p$, meaning that $p_i = 0$ implies $q_i = 0$. Note that if both $p$ and $q$ are probability vectors, meaning that $\sum_i p_i = \sum_i q_i = 1$, we obtain

$$D_{\mathrm{KL}}(p||q) := \sum_{i=1}^d p_i \log \frac{p_i}{q_i}. \tag{37}$$

**Information Geometry of Policy Optimization**  Among the most successful policy optimization schemes are natural policy gradient (NPG) methods or variants thereof like trust-region and proximal policy optimization (TRPO and PPO, respectively). These methods assume a convex geometry and corresponding Bregman divergences in the state-action polytope, where we refer to (Neu et al., 2017; Müller & Montúfar, 2023) for a more detailed discussion.

In general, a trust region update is defined as

$$\pi_{k+1} \in \arg\max_{\pi \in \Pi} \mathbb{A}_r^{\pi_k}(\pi) \quad \text{sbj. to } D_\Phi(d_{\pi_k}||d_\pi) \leq \delta, \tag{38}$$

where $D_\Phi \colon \mathscr{D} \times \mathscr{D} \to \mathbb{R}$ is a Bregman divergence induced by a suitably convex function $\Phi \colon \mathrm{int}(\mathscr{D}) \to \mathbb{R}$. The functional

$$\mathbb{A}_r^{\pi_k}(\pi) = \mathbb{E}_{s \sim d_{\pi_k}, a \sim \pi_\theta(\cdot|s)}\big[A_r^{\pi_k}(s, a)\big], \tag{39}$$

as introduced in (Kakade & Langford, 2002), is called the *policy advantage*. As a loss function, it is also known as the surrogate advantage (Schulman et al., 2017a), since we can interpret $\mathbb{A}$ as a surrogate optimization objective of the return. In particular, it holds for a parameterized policy $\pi_\theta$, that $\nabla_\theta \mathbb{A}_r^{\pi_{\theta_k}}(\pi_\theta)|_{\theta=\theta_k} = \nabla_\theta V_r(\theta_k)$, see (Kakade & Langford, 2002; Schulman et al., 2017a). TRPO and the original NPG assume the same geometry (Kakade, 2001; Schulman et al., 2017a), since they employ an identical Bregman divergence

$$D_{\mathrm{K}}(d_{\pi_1}||d_{\pi_2}) := \sum_{s,a} d_{\pi_1}(s, a) \log \frac{\pi_1(a|s)}{\pi_2(a|s)} = \sum_s d_{\pi_1}(s) D_{\mathrm{KL}}(\pi_1(\cdot|s)||\pi_2(\cdot|s)).$$

We refer to $D_{\mathrm{K}}$ as the Kakade divergence and informally write $D_{\mathrm{K}}(\pi_1, \pi_2) := D_{\mathrm{K}}(d_{\pi_1}, d_{\pi_2})$. This divergence can be shown to be the Bregman divergence induced by the negative conditional entropy

$$\Phi_{\mathrm{K}}(d_\pi) := \sum_{s,a} d_\pi(s, a) \log \pi(a|s), \tag{40}$$

see (Neu et al., 2017). It is well known that with a parameterized policy $\pi_\theta$, a linear approximation of $\mathbb{A}$ and a quadratic approximation of the Bregman divergence $D_{\mathrm{K}}$ at $\theta$, one obtains the *natural policy gradient* step given by

$$\theta_{k+1} = \theta_k + \epsilon_k G_{\mathrm{K}}(\theta_k)^+ \nabla R(\theta_k), \tag{41}$$

where $G_{\mathrm{K}}(\theta)^+$ denotes a pseudo-inverse of the Gramian matrix with entries equal to the state-averaged Fisher-information matrix of the policy

$$G_{\mathrm{K}}(\theta)_{ij} := \mathbb{E}_{s \sim d_{\pi_\theta}} \left[ \sum_a \frac{\partial_{\theta_i} \pi_\theta(a|s) \partial_{\theta_j} \pi_\theta(a|s)}{\pi_\theta(a|s)} \right] \tag{42}$$

$$= \mathbb{E}_{d_{\pi_\theta}} [\partial_{\theta_i} \log \pi_\theta(a|s) \partial_{\theta_j} \log \pi_\theta(a|s)], \tag{43}$$

where we refer to (Schulman et al., 2017a) for a more detailed discussion.

Consider a convex potential $\Phi \colon \mathscr{D} \to \mathbb{R}$ or $\Phi \colon \mathscr{D}_{\mathrm{safe}} \to \mathbb{R}$ and the TRPO update

$$\theta_{k+1} \in \arg\max \mathbb{A}_r^{\pi_{\theta_k}}(\pi_\theta) \quad \text{sbj. to } D_\Phi(d_{\theta_k}||d_\theta) \leq \epsilon. \tag{44}$$

In practice, one uses a linear approximation of $\mathbb{A}_r^{\pi_{\theta_k}}(\pi_\theta)$ and a quadratic approximation of $D_\Phi$ to compute the TRPO update. This gives the following approximation of TRPO

$$\theta_{k+1} \in \arg\max_\theta \nabla_\theta \mathbb{A}_r^{\theta_k}(\theta)|_{\theta=\theta_k} \cdot (\theta - \theta_k) \quad \text{sbj. to } \|\theta - \theta_k\|_{G(\theta_k)}^2 \leq \epsilon, \tag{45}$$

where

$$G(\theta)_{ij} = \partial_{\theta_i} d_\theta^\top \nabla^2 \Phi(d_\theta) \partial_{\theta_j} d_\theta. \tag{46}$$

Note that by the policy gradient theorem, it holds that

$$\nabla_\theta \mathbb{A}_r^{\theta_k}(\theta)|_{\theta=\theta_k} = \nabla V_r(\theta_k). \tag{47}$$

Thus, the approximate TRPO update is equivalent to

$$\theta_{k+1} = \theta_k + \epsilon_k G(\theta_k)^+ \nabla V_r(\theta), \tag{48}$$

where

$$\epsilon_k = \frac{\sqrt{\epsilon}}{\|G(\theta_k)^+ \nabla V_r(\theta_k)\|_{G(\theta_k)}}. \tag{49}$$

Hence, the approximation TRPO update corresponds to a natural policy gradient update with an adaptively chosen step size.

## B. Details on the Safe Geometry for CMDPs

### B.1. Safe Trust Regions

The safe mirror function for a single constraint is given by

$$\Phi_{\mathrm{C}}(d) := \Phi_{\mathrm{K}}(d) + \sum_{i=1}^m \beta\,\phi(b - c^\top d), \tag{50}$$

and the resulting Bregman divergence

$$D_{\mathrm{C}}(d_1||d_2) = \Phi_{\mathrm{C}}(d_1) - \Phi_{\mathrm{C}}(d_2) - \langle \nabla\Phi_{\mathrm{C}}(d_2), d_1 - d_2 \rangle. \tag{51}$$

is a linear operator in $\Phi$, hence

$$D_{\Phi(d)+\beta\phi(b-c^\top d)}(d_1||d_2) = D_{\Phi_{\mathrm{K}}}(d_1||d_2) + \beta D_\phi(d_1||d_2), \tag{52}$$

where

$$D_\phi(d_1||d_2) = \phi(b - c^\top d_1) - \phi(b - c^\top d_2) - \langle \nabla\phi(b - c^\top d_2), d_1 - d_2 \rangle \tag{53}$$

$$= \phi(b - c^\top d_1) - \phi(b - c^\top d_2) - \phi'(b - c^\top d_2)(c^\top d_1 - c^\top d_2). \tag{54}$$

$$= \phi(b - V_c(\pi_1)) - \phi(b - V_c(\pi_2)) + \phi'(b - V_c(\pi_2))(V_c(\pi_1) - V_c(\pi_2)). \tag{55}$$

The last expression can be interpreted as the one-dimensional Bregman divergence $D_\phi(b - V_c(\pi)||b - V_c(\pi_k))$, which is a (strictly) convex function in $V_c(\pi)$ for fixed $\pi_k$ if $\phi$ is (strictly) convex.

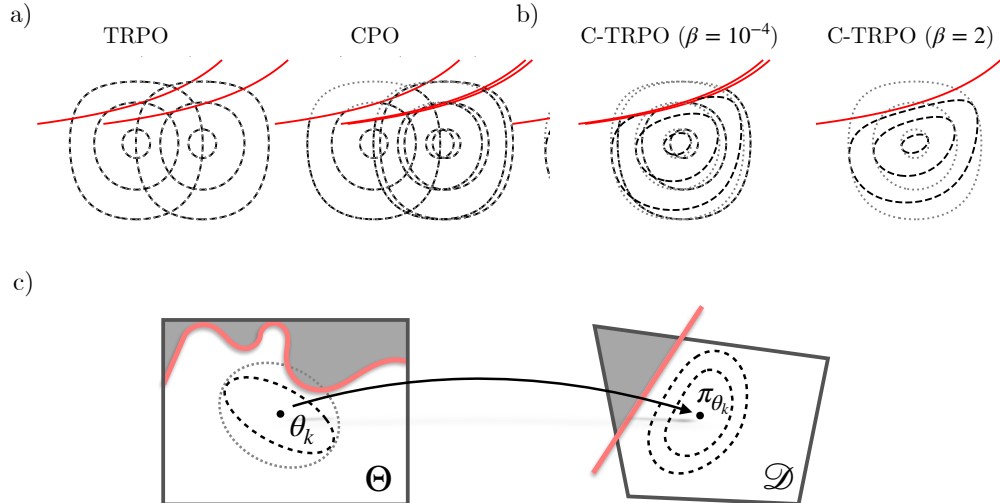

Figure 5: Illustration of policy divergences (dashed) close to the constraint (red). a) TRPO (dotted for reference) and CPO. b) C-TRPO's divergence depends on the hyper-parameter $\beta$, which modulates the strength of the barrier towards the constraint surface. For $\beta \searrow 0$ we obtain an update equivalent to CPO, and more conservative updates for larger values ($\beta = 2$). The plots were generated with the toy MDP in Figure 2. c) Shown are the quadratic approximations of the divergence in parameter space, which is obtained by mapping the policy onto its occupancy measure, where a safe geometry can be defined using standard tools from convex optimization (safe region in white).

## B.2. Details on C-TRPO

### B.2.1. SURROGATE DIVERGENCE

In practice, the exact constrained KL-Divergence $D_C$ cannot be evaluated, because it depends on the cost-return of the optimized policy $V_c(\pi)$. Therefore, we use the surrogate divergence

$$\bar{D}_\phi(\pi_\theta||\pi_{\theta_k}) = \phi(b - V_c^{\pi_k} - \mathbb{A}_c^{\pi_k}(\pi)) - \phi(b - V_c^{\pi_k}) + \phi'(b - V_c^{\pi_k})\mathbb{A}_c^{\pi_k}(\pi) \tag{56}$$

which is obtained by the substitution $V_c(\pi) - V_c^{\pi_k} \to \mathbb{A}_c^{\pi_k}(\pi)$ in $D_\phi$.

When we center this divergence around policy $\pi_k$ and keep this policy fixed, it becomes a function of the policy cost advantage.

$$\begin{aligned}
\bar{D}_\phi(\pi_\theta||\pi_{\theta_k}) &= \phi(b - V_c^{\pi_k} - \mathbb{A}_c^{\pi_k}(\pi)) - \phi(b - V_c^{\pi_k}) + \phi'(b - V_c^{\pi_k})\mathbb{A}_c^{\pi_k}(\pi) \\
&= \phi(\delta_b - \mathbb{A}_c^{\pi_k}(\pi)) - \phi(\delta_b) + \phi'(\delta_b)\mathbb{A}_c^{\pi_k}(\pi) \\
&= \Psi(\mathbb{A}_c^{\pi_k}).
\end{aligned}$$

Note that $\bar{D}_\phi(\pi_\theta||\pi_{\theta_k}) = \Psi(\mathbb{A}_c^{\pi_k}(\pi))$, where $\Psi(x) = \phi(\delta_b - x) - \phi(\delta_b) - \phi'(\delta_b) \cdot x$ is a (strictly) convex function if $\phi$ is (strictly) convex, since it is equivalent to the one-dimensional Bregman divergence $D_\phi(\delta_b - x||\delta_b)$ on the domain of $\phi(b - x)$, see Figure 6.

**Example B.1.** The function $\phi(x) = x\log(x)$ induces the divergence

$$\bar{D}_\phi(\pi_\theta||\pi_{\theta_k}) = \mathbb{A}_c^{\pi_k}(\pi_\theta) - (\delta_b - \mathbb{A}_c^{\pi_k}(\pi_\theta))\log\left(\frac{\delta_b}{\delta_b - \mathbb{A}_c^{\pi_k}(\pi_\theta)}\right). \tag{57}$$

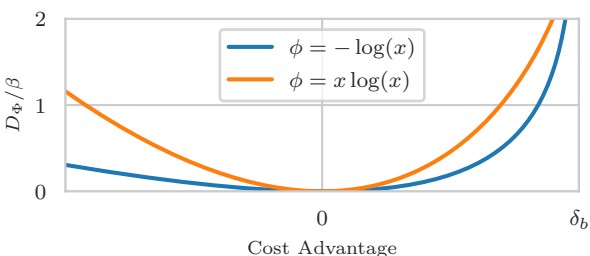

Figure 6: The surrogate Constrained KL-Divergence as a function of the policy cost advantage.

### B.2.2. ESTIMATION

In the practical implementation, the expected KL-divergence between the policy of the previous iteration, $\pi_k$, and the proposal policy $\pi$ is estimated from state samples $s_i$ by running $\pi_k$ in the environment

$$\sum_s d_{\pi_k}(s) D_{\mathrm{KL}}(\pi(\cdot|s)||\pi_k(\cdot|s)) \approx 1/N \sum_{i=0}^{N-1} D_{\mathrm{KL}}(\pi(\cdot|s_i)||\pi_k(\cdot|s_i)) \tag{58}$$

where $D_{\mathrm{KL}}$ can be computed in closed form for Gaussian policies, where $N$ is the batch size.

For the constraint term, we estimate $\delta_b$ from trajectory samples, as well as the policy cost advantage

$$\mathbb{A}_c^{\pi_k}(\pi) \approx \hat{\mathbb{A}} = \frac{1}{N} \sum_{i=0}^{N-1} \frac{\pi(a_i|s_i)}{\pi_k(a_i|s_i)} \hat{A}_i^{\pi_k} \tag{59}$$

where $\hat{A}_i^{\pi_k}$ is the GAE-$\lambda$ estimate (Schulman et al., 2018) of the advantage function of the cost. For any suitable $\phi$, the resulting divergence estimate is

$$\hat{D}_\phi = \phi(\delta_b - \hat{\mathbb{A}}) - \phi(\delta_b) - \phi'(\delta_b)\hat{\mathbb{A}} \tag{60}$$

and for the specific choice $\phi(x) = x\log(x)$

$$\hat{D}_\phi = \hat{\mathbb{A}} - (\delta_b - \hat{\mathbb{A}}) \log\left(\frac{\delta_b}{\delta_b - \hat{\mathbb{A}}}\right). \tag{61}$$

### B.2.3. DETAILS ON C-NPG

In showing that TRPO with quadratic approximation agrees with a natural gradient step, see Appendix A, we have used that $\nabla_\theta \mathbb{A}_r^{\theta_k}(\theta)|_{\theta=\theta_k} = \nabla V_r(\theta_k)$, which holds although $\mathbb{A}_r$ is only a proxy of $V_r$. We now provide a similar property for the quadratic approximation of the surrogate divergences $\bar{D}_C$.

**Proposition B.2.** *For any parameter $\theta$ with $\pi_\theta \in \Pi_{\mathrm{safe}}$ it holds that*

$$\nabla_\theta^2 \bar{D}_\phi(\theta||\hat{\theta})|_{\theta=\hat{\theta}} = \nabla_\theta^2 D_\phi(\theta||\hat{\theta})|_{\theta=\hat{\theta}} \tag{62}$$

*and hence*

$$\nabla_\theta^2 \bar{D}_{\mathrm{KL}}(\theta||\hat{\theta})|_{\theta=\hat{\theta}} + \beta \nabla_\theta^2 \bar{D}_\phi(\theta||\hat{\theta})|_{\theta=\hat{\theta}} = G_C(\hat{\theta}) \tag{63}$$

*where $G_C(\theta)$ denotes the Gramian matrix of C-NPG with entries*

$$G_C(\theta)_{ij} = \partial_{\theta_i} d_\theta^\top \nabla^2 \Phi_C(\theta) \partial_{\theta_j} d_\theta. \tag{64}$$

*Proof.* Let $\bar{H}_{\mathrm{KL}}(\theta) = \nabla_\theta^2 \bar{D}_{\mathrm{KL}}(\theta||\hat{\theta})|_{\theta=\hat{\theta}}$ and $\bar{H}_\phi(\theta) = \nabla_\theta^2 \bar{D}_\phi(\theta||\hat{\theta})|_{\theta=\hat{\theta}}$. One can show that $\bar{H}_{\mathrm{KL}} = G_{\mathrm{K}}(\theta)$ (Schulman et al., 2017a). Further, we have

$$
\begin{aligned}
\bar{H}_\phi(\theta) &= \nabla_\theta \mathbb{A}_c^{\pi_k}(\theta) \Psi''(\mathbb{A}_c^{\pi_k}(\theta)) \nabla_\theta \mathbb{A}_c^{\pi_k}(\theta)^\top + \Psi'(\mathbb{A}_c^{\pi_k}(\theta)) \nabla_\theta^2 \mathbb{A}_c^{\pi_k}(\theta) \\
&\overset{a)}{=} \nabla_\theta \mathbb{A}_c^{\pi_k}(\theta) \Psi''(\mathbb{A}_c^{\pi_k}(\theta)) \nabla_\theta \mathbb{A}_c^{\pi_k}(\theta)^\top \\
&\overset{b)}{=} \nabla_\theta \mathbb{A}_c^{\pi_k}(\theta) \phi''(b - V_c^{\pi_k}(\theta)) \nabla_\theta \mathbb{A}_c^{\pi_k}(\theta)^\top \\
&= \nabla_\theta V_c^{\pi_k}(\theta) \phi''(b - V_c^{\pi_k}(\theta)) \nabla_\theta V_c^{\pi_k}(\theta)^\top,
\end{aligned}
$$

where a) follows from $\Psi'(\mathbb{A}_{\underline{c}}^{\pi_k}(\theta)) = 0$ since $\Psi(0) = 0$, $\Psi \geq 0$ and $\mathbb{A}_c^{\hat{\theta}}(\theta)|_{\theta=\hat{\theta}} = 0$. Further, b) follows because $\Psi''(x)|_{x=0} = \phi''(\delta_b)$. Thus, $\bar{H}_\phi$ is equivalent to the Gramian

$$
\begin{aligned}
G_{\mathrm{C}}(\theta)_{ij} &:= \partial_{\theta_i} d_\theta^\top \nabla^2 \Phi_{\mathrm{C}}(\theta) \partial_{\theta_j} d_\theta && (65) \\
&= G_{\mathrm{K}}(\theta)_{ij} + \beta \phi''(b - c_k^\top d_\theta) \partial_{\theta_i} d_\theta^\top cc^\top \partial_{\theta_i} d_\theta && (66) \\
&= \bar{H}_{\mathrm{KL}} + \beta \nabla_\theta V_c(\theta) \phi''(b - V_c(\theta)) \nabla_\theta V_c(\theta)^\top, && (67) \\
&= \bar{H}_{\mathrm{KL}} + \beta \bar{H}_\phi. && (68)
\end{aligned}
$$

Again, for multiple constraints, the statement follows analogously. □

In particular, this shows that the C-TRPO update can be interpreted as a natural policy gradient step with an adaptive step size and that the updates with $D_{\mathrm{C}}$ and $\bar{D}_{\mathrm{C}}$ are equivalent if we use a quadratic approximation for both, justifying $\bar{D}_{\mathrm{C}}$ as a surrogate for $D_{\mathrm{C}}$.

## B.3. Beyond finite MDPs

For the sake of simplicity and as this is required for our theoretical analysis, we have introduced C-TRPO only for finite MDPs. However, C-TRPO can also be used for problems with continuous state and action spaces as we discuss here. In this case, the state-action and state distributions are defined as

$$
d_\pi(S \times A) := (1 - \gamma) \sum_{t=0}^\infty \gamma^t \mathbb{P}_\pi(s_t \in S, a_t \in A) \quad \text{and}
$$

$$
d_\pi(S) := (1 - \gamma) \sum_{t=0}^\infty \gamma^t \mathbb{P}_\pi(s_t \in S)
$$

for any measurable subsets $A \subseteq \mathcal{A}$ and $S \subseteq \mathcal{S}$. Further, the Kakade divergence is then given by

$$
D_{\mathrm{K}}(d^{\pi_1}||d^{\pi_2}) := \mathbb{E}_{s \sim d^{\pi_1}} \left[ D_{\mathrm{KL}}(\pi_1(\cdot|s)||\pi_2(\cdot|s)) \right], \tag{69}
$$

which is well defined if $\pi_1(\cdot|s)$ is absolutely continuous with respect to $\pi_2(\cdot|s)$ for $d^{\pi_1}$ almost all $s \in \mathcal{S}$. The Bregman divergence that C-TRPO builds on is – just as in the finite case – given by

$$
D_{\mathrm{C}}(d_1||d_2) = D_{\mathrm{K}}(d_1||d_2) + \sum_{i=1}^m \beta_i D_{\phi_i}(d_1||d_2), \tag{70}
$$

where

$$
D_{\phi_i}(d_1||d_2) = \phi(b_i - V_{c_i}(\pi_1)) - \phi(b_i - V_{c_i}(\pi_2)) + \phi'(b_i - V_{c_i}(\pi_2))(V_{c_i}(\pi_1) - V_{c_i}(\pi_2)). \tag{71}
$$

Like in the finite case, the policy advantage is defined as

$$
\mathbb{A}_r^{\pi_k}(\pi) = \mathbb{E}_{s, a \sim d_{\pi_k}} \left[ \frac{\pi(a|s)}{\pi_k(a|s)} A_r^{\pi_k}(s, a) \right], \tag{72}
$$

where $A_r^\pi(s,a) = Q^\pi(s,a) - V^\pi(s)$ denotes the advantage function, which is defined analoguously to the finite case. Now, the plain trust region update is given by

$$\theta_{k+1} \in \arg\max_\theta \mathbb{A}_r^{\pi_k}(\pi) \quad \text{sbj. to } D_{\mathrm{C}}(d_{\pi_k}||d_\pi) \le \delta. \tag{73}$$

Just like in the finite case, we use a surrogate divergence $\bar{D}_{\mathrm{C}}$ and obtain the formulation of C-TRPO

$$\pi_{k+1} = \arg\max_{\pi \in \Pi} \mathbb{A}_r^{\pi_k}(\pi) \quad \text{sbj. to } \bar{D}_{\mathrm{C}}(\pi||\pi_k) \le \delta. \tag{74}$$

Here, the differences to $D_{\mathrm{C}}$ are that we use samples from the state distribution $d^{\pi_k}$ and use a surrogate for the cost advantage to estimate the divergence $D_{\phi_i}$ as described in Section 3.2. Further, we use a parametric policy model $\pi_\theta$ and a linear approximation of $\mathbb{A}^{\pi_k}$ as well as quadratic approximation of $\bar{D}_{\mathrm{C}}(\pi||\pi_k)$ for our practical implementation.

**Expression for Gaussian policies**  We test C-TRPO in various control tasks where we use Gaussian policies. More precisely, the state and action space consist of Euclidean spaces $\mathcal{S} = \mathbb{R}^{d_s}$ and $\mathcal{A} = R^{d_a}$. Then, we consider a policy network $\mu_\theta \colon \mathcal{S} \to \mathcal{A}$, which predicts the mean action and assume parameterized but state independent diagonal Gaussian noise, meaning that $\pi_\theta(\cdot|s) = \mathcal{N}(\mu_\theta(s), \Sigma_\theta)$, where $\Sigma_\theta$ is diagonal. Consequently, we can use a closed-form expression for the KL divergence as

$$D_{\mathrm{KL}}(\pi_{\theta_1}(\cdot|s)||\pi_{\theta_2}(\cdot|s)) = \frac{1}{2}\left(\mathrm{tr}\left(\Sigma_{\theta_2}^{-1}\Sigma_{\theta_1}\right) - d_a + \|\mu_{\theta_1}(s) - \mu_{\theta_2}(s)\|_{\Sigma_{\theta_2}^{-1}}^2 + \ln\left(\frac{\det \Sigma_{\theta_2}}{\det \Sigma_{\theta_1}}\right)\right),$$

see (Zhang et al., 2024b).

## C. Proofs of Section 4

### C.1. Proofs of Section 4.1

Our theoretical analysis of C-TRPO is built on the following bounds on the performance difference of two policies.

**Theorem C.1** (Performance Difference, (Achiam et al., 2017)). *For any function $f(s,a)$, the following bounds hold*

$$V_f(\pi_1) - V_f(\pi_2) \lesseqgtr \mathbb{A}_f^{\pi_2}(\pi_1) \pm \frac{2\gamma\epsilon_f}{(1-\gamma)}\sqrt{\frac{1}{2}\mathbb{E}_{s\sim d_{\pi_2}}D_{\mathrm{KL}}(\pi_1(\cdot|s)||\pi_2(\cdot|s))} \tag{75}$$

*where $\epsilon_f = \max_s |\mathbb{E}_{a\sim\pi_1}A_f^{\pi_2}(s,a)|$.*

Theorem C.1 can be interpreted as a bound on the error incurred by replacing the difference in returns $V_f(\pi_1) - V_f(\pi)$ of any state-action function by its policy advantage $\mathbb{A}_f^{\pi_2}(\pi_1)$.

**Proposition 4.1** (C-TRPO reward update). *Set $\epsilon_r = \max_s |\mathbb{E}_{a\sim\pi_{k+1}}A_r^{\pi_k}(s,a)|$. The expected reward of a policy updated with C-TRPO is bounded from below by*

$$V_r(\pi_{k+1}) \ge V_r(\pi_k) - \frac{\sqrt{2\delta}\gamma\epsilon_r}{1-\gamma}. \tag{23}$$

*Proof.* It follows from the lower bound in Theorem C.1 that

$$V_r(\pi_{k+1}) - V_r(\pi_k) \ge \mathbb{A}_r^{\pi_k}(\pi_{k+1}) - \frac{\gamma\epsilon_r}{(1-\gamma)}\sqrt{2\bar{D}_{\mathrm{C}}(\pi_{k+1}||\pi_k)} \tag{76}$$

where we choose $f = r$. The bound holds because $\bar{D}_\phi \ge 0$, and thus $\bar{D}_{\mathrm{C}} \ge \mathbb{E}D_{\mathrm{KL}}$. Further, $\delta \ge D_{\mathrm{C}}$ and $\mathbb{A}_r^{\pi_k}(\pi_{k+1}) \ge 0$ by the update equation, which concludes the proof. See Appendix C.3 for a more detailed discussion. $\square$

**Proposition 4.2.** *The approximate C-TRPO update approaches the CPO update in the limit as $\beta \searrow 0$.*

*Proof.* Let us fix a strictly safe policy $\pi_0 \in \text{int}(\Pi_{\text{safe}})$. In both cases, we approximate the expected cost of a policy using $V_c(\pi) \approx V_c(\pi_0) + \mathbb{A}_c^{\pi_0}(\pi)$, which is off by the advantage mismatch term in Proposition 4.1. Hence, we maximize the surrogate of the expected value $\mathbb{A}_r^{\pi_0}(\pi)$ over the regions

$$P_{\text{CPO}} := \{\pi \in \Pi : \bar{D}_{\text{K}}(\pi, \pi_0) \leq \delta, \ V_c(\pi_0) + \mathbb{A}_c^{\pi_0}(\pi) \leq b\}$$

in the case of CPO, and

$$P_\beta := \{\pi \in \Pi : \bar{D}_{\text{C}}(\pi, \pi_0) \leq \delta\},$$

with C-TRPO for some $\beta > 0$. Note that

$$\bar{D}_{\text{C}}(\pi, \pi_0) = \bar{D}_{\text{K}}(\pi, \pi_0) + \beta \Psi(\mathbb{A}_c^{\pi_0}(\pi)), \tag{77}$$

and $\Psi : (-\infty, \delta_b) \to (0, +\infty)$ and $\Psi(t) \to +\infty$ for $t \nearrow \delta_b$, where $\delta_b = b - V_c(\pi_0)$. Denote the corresponding updates by $\hat{\pi}_{\text{CPO}}$ and the C-TRPO update by $\hat{\pi}_\beta$. Note that we have $P_\beta \subseteq P_{\beta'} \subseteq P_{\text{CPO}}$ for $\beta \geq \beta'$. Further, we have

$$\bigcup_{\beta > 0} P_\beta = \{\pi \in P : D_{\text{K}}(\pi, \pi_0) < \delta, V_c(\pi_0) + \mathbb{A}_c^{\pi_0}(\pi) < b\}.$$

Hence, the trust regions $P_\beta$ grow for $\beta \searrow 0$ and fill the interior of the trust region $P_{\text{CPO}}$. $\qquad\square$

*Remark* C.2. Intuitively, one could repeatedly solve the C-TRPO problem with successively smaller values of $\beta$, which would be similar to solving CPO with the interior point method using $\Psi$ as the barrier function.

**Proposition 4.3** (C-TRPO worst-case constraint violation). *Let* $\bar{D}_{\text{C}}(\pi_{k+1}||\pi_k) \leq \delta$ *with* $\delta > 0$ *and set* $\epsilon_c = \max_s |\mathbb{E}_{a \sim \pi_{k+1}} A_c^{\pi_k}(s, a)|$. *It holds that*

$$V_c(\pi_{k+1}) \leq V_c(\pi_k) + \mathbb{A}_c^{\pi_k}(\pi_{k+1}) + \frac{\sqrt{2\delta(\beta)}\gamma\epsilon_c}{1-\gamma}, \tag{24}$$

*where* $\delta(\beta) = \delta - \beta \bar{D}_\phi(\pi_{k+1}, \pi_k) \leq \delta$ *is decreasing in* $\beta > 0$, $\lim_{\beta \to 0} \delta(\beta) = \delta$, *and* $\delta(\beta) \to 0$ *for* $\beta \to \delta \bar{D}_{\text{C}}(\pi_{k+1}||\pi_k)/\bar{D}_\phi(\pi_{k+1}, \pi_k)$.

*Proof.* Setting $f = c$ in the upper bound from Theorem C.1 we obtain

$$V_c(\pi_{k+1}) \leq V_c(\pi_k) + \mathbb{A}_c^{\pi_k}(\pi_{k+1}) \pm \frac{2\gamma\epsilon_c}{(1-\gamma)} \sqrt{\frac{1}{2}\mathbb{E}_{s \sim d_{\pi_k}} D_{\text{KL}}(\pi_{k+1}(\cdot|s)||\pi_k(\cdot|s))}$$

Note now that we have

$$\mathbb{E}_{s \sim d_{\pi_k}} D_{\text{KL}}(\pi_{k+1}(\cdot|s)||\pi_k(\cdot|s)) = \bar{D}_{\text{C}}(\pi_{k+1}||\pi_k) - \beta \bar{D}_\phi(\pi_{k+1}, \pi_k) \leq \delta - \beta \bar{D}_\phi(\pi_{k+1}, \pi_k) = \delta(\beta).$$

$$\qquad\square$$

### C.2. Details on the results in Section 4.2

Recall that we study the natural policy gradient flow

$$\partial_t \theta_t = G_{\text{C}}(\theta_t)^+ \nabla V_r(\theta_t), \tag{78}$$

where $G_{\text{C}}(\theta)^+$ denotes a pseudo-inverse of $G_{\text{C}}(\theta)$ with entries

$$G_{\text{C}}(\theta)_{ij} := \partial_{\theta_i} d_\theta^\top \nabla^2 \Phi_{\text{C}}(d_\theta) \partial_{\theta_j} d_\theta = G_{\text{K}}(\theta)_{ij} + \sum_k \beta_k \phi''(b_k - c_k^\top d_\theta) \partial_{\theta_i} d_\theta^\top c_k c_k^\top \partial_{\theta_i} d_\theta. \tag{79}$$

and $\theta \mapsto \pi_\theta$ is a differentiable policy parametrization.

Moreover, we assume that $\theta \mapsto \pi_\theta$ is regular, that it is surjective and the Jacobian is of maximal rank everywhere. This assumption implies overparametrization but is satisfied for common models like tabular softmax, tabular escort, or expressive log-linear policy parameterizations (Agarwal et al., 2021a; Mei et al., 2020a; Müller & Montúfar, 2023).

We denote the set of safe parameters by $\Theta_{\text{safe}} := \{\theta \in \mathbb{R}^p : \pi_\theta \in \Pi_{\text{safe}}\}$, which is non-convex in general and say that $\Theta_{\text{safe}}$ is *invariant* under Equation (25) if $\theta_0 \in \Theta_{\text{safe}}$ implies $\theta_t \in \Theta_{\text{safe}}$ for all $t$. Invariance is associated with safe control during optimization and is typically achieved via control barrier function methods (Ames et al., 2017; Cheng et al., 2019). We study the evolution of the state-action distributions $d_t = d^{\pi_{\theta_t}}$ as this allows us to employ the linear programming formulation of CMPDs and we obtain the following convergence guarantees.

**Theorem 4.4** (Safety during training). *Assume that $\phi \colon \mathbb{R}_{>0} \to \mathbb{R}$ satisfies $\phi'(x) \to +\infty$ for $x \searrow 0$ and consider a regular policy parameterization. Then the set $\Theta_{\text{safe}}$ is invariant under Equation (25).*

*Proof.* Consider a solution $(\theta_t)_{t>0}$ of Equation (78). As the mapping $\pi \mapsto d^\pi$ is a diffeomorphism (Müller & Montúfar, 2023) the parameterization $\Theta_{\text{safe}} \to \mathscr{D}_{\text{safe}}, \theta \mapsto d^{\pi_\theta}$ is surjective and has a Jacobian of maximal rank everywhere. As $G_{\text{C}}(\theta)_{ij} = \partial_{\theta_i} d_\theta \nabla \Phi_{\text{C}} \partial_{\theta_i} d_\theta$ this implies that the state-action distributions $d_t = d^{\pi_{\theta_t}}$ solve the Hessian gradient flow with Legendre-type function $\Phi_{\text{C}}$ and the linear objective $d \mapsto r^\top d$, see (Amari, 2016; van Oostrum et al., 2023; Müller & Montúfar, 2023) for a more detailed discussion. It suffices to study the gradient flow in the space of state-action distributions $d_t$. It is easily checked that $\Phi_{\text{C}}$ is a Legendre-type function for the convex domain $\mathscr{D}_{\text{C}}$, meaning that it satisfies $\|\nabla \Phi(d_n)\| \to +\infty$ for $d_n \to d \in \partial \mathscr{D}_{\text{safe}}$. Since the objective is linear, it follows from the general theory of Hessian gradient flows of convex programs that the flow is well posed, see (Alvarez et al., 2004; Müller & Montúfar, 2023). $\qquad\square$

**Theorem 4.5.** *Assume that $\phi'(x) \to +\infty$ for $x \searrow 0$, set $V_{r,\text{C}}^\star := \max_{\pi \in \Pi_{\text{safe}}} V_r(\pi)$ and denote the set of optimal constrained policies by $\Pi_{\text{safe}}^\star = \{\pi \in \Pi_{\text{safe}} : V_r(\pi) = V_{r,\text{C}}^\star\}$, consider a regular policy parametrization and let $(\theta_t)_{t \geq 0}$ solve Equation (25). It holds that $V_r(\pi_{\theta_t}) \to V_{r,\text{C}}^\star$ and*

$$\lim_{t \to +\infty} \pi_t = \pi_{\text{safe}}^\star = \arg\min\{D_{\text{C}}(\pi^\star, \pi_0) : \pi^\star \in \Pi_{\text{safe}}^\star\}. \tag{26}$$

*Proof.* Just like in the proof of Theorem 4.4 we see that $d_t = d^{\pi_{\theta_t}}$ solves the Hessian gradient flow with respect to the Legendre type function $\Phi_{\text{C}}$. Now the claims regarding convergence and the identification of the limit $\lim_{t \to +\infty} \pi_{\theta_t}$ follows from the general theory of Hessian gradient flows, see (Alvarez et al., 2004; Müller et al., 2024). $\qquad\square$

## C.3. Performance improvement bounds and choice of divergence

In a series of works (Kakade & Langford, 2002; Pirotta et al., 2013; Schulman et al., 2017a; Achiam et al., 2017), the following bound on policy performance difference between two policies has been established.

$$V_f(\pi') - V_f(\pi) \lesseqgtr \mathbb{A}_f^{\pi'}(\pi) \pm \frac{2\gamma \epsilon_f}{(1-\gamma)} \mathbb{E}_{s \sim d_\pi} D_{\text{TV}}(\pi' \| \pi)(s) \tag{80}$$

where $D_{\text{TV}}$ is the Total Variation Distance. Furthermore, by Pinsker's inequality, we have that

$$D_{\text{TV}}(\pi' \| \pi) \leq \sqrt{\frac{1}{2} D_{\text{KL}}(\pi' \| \pi)}, \tag{81}$$

and by Jensen's inequality

$$\mathbb{E}_{s \sim d_\pi} D_{\text{TV}}(\pi' \| \pi)(s) \leq \sqrt{\frac{1}{2} \mathbb{E}_{s \sim d_\pi} D_{\text{KL}}(\pi' \| \pi)(s)}, \tag{82}$$

It follows that we can not only substitute the KL-divergence into the bound but any divergence

$$D_\Phi(d_\pi' \| d_\pi) \geq \mathbb{E}_{s \sim d_\pi} D_{\text{KL}}(\pi' \| \pi)(s) \tag{83}$$

can be substituted, and still retains TRPO's and CPO's update guarantees.

## C.4. Comparison with CPO

In the approximate case of C-TRPO and CPO, where the reward is approximated linearly, and the trust region quadratically, the constraints differ in that C-TRPO's constraint is

$$(\theta - \theta_k)(\bar{H}_{\text{KL}}(\theta) + \beta \bar{H}_\phi(\theta))(\theta - \theta_k) < \delta$$

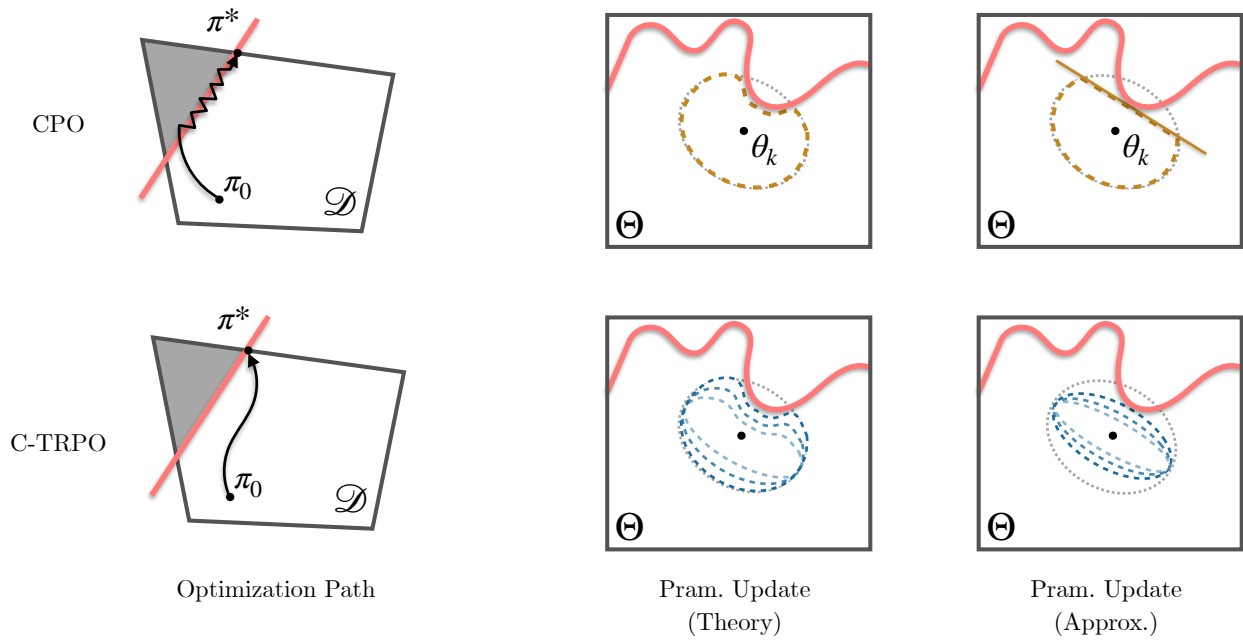

Figure 7: Pictorial illustration of conceptual and practical differences between CPO and C-TRPO. The local approximation of C-TRPO's trust region results in a single quadratic constraint, which is compressed in the direction of the closest cost surface, depending on the hyper-parameter $\beta$ (blue dashed lines on the right). This is in contrast to CPO, where the local approximation of the update results in a quadratic constraint which is not affected by the cost, and a linear constraint which only takes effect upon contact with the cost surface. Intuitively, this results in a smoother optimization path for C-TRPO that remains on the interior of the safe policy space for longer.

whereas CPO's is

$$(\theta - \theta_k)\bar{H}_{\mathrm{KL}}(\theta)(\theta - \theta_k) < \delta \text{ and } V_c^{\theta_k} + (\nabla_\theta \mathbb{A}_c^{\theta_k}(\theta))^\top (\theta - \theta_k) \leq b.$$

Figure 7 illustrates the differences between CPO and C-TRPO.

# D. Additional Experiments

## D.1. Effect of hyper-parameters

To better understand the effects of the two hyperparameters $\beta$ and $b_\mathrm{H}$, we observe how they change the training dynamics through the example of the *AntVelocity* environment.

The safety parameter $\beta$ modulates the stringency with which C-TRPO satisfies the constraint, without limiting the expected return for values up to $\beta = 1$, see Figure 8. For higher values, the expected return starts to degrade, partly due to $\bar{D}_\phi$ being relatively noisy compared to $\bar{D}_\mathrm{KL}$ and thus we recommend the choice $\beta = 1$.

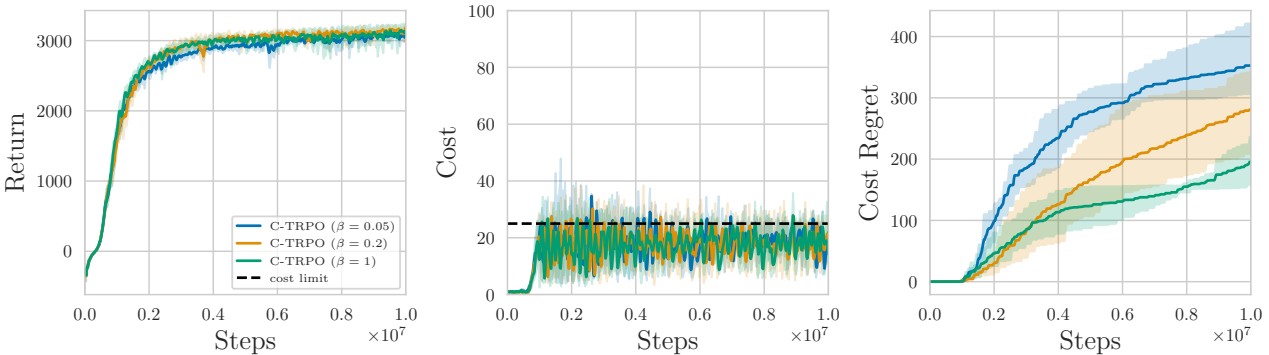

Figure 8: Influence of $\beta$ on C-TRPO's performance.

Finally, employing a hysteresis fraction $0 < b_\mathrm{H} < b$ seems beneficial, possibly because it leads the iterate away from the boundary of the safe set, and because divergence estimates tend to be more reliable for strictly safe policies. The effect of the choice of $b_\mathrm{H}$ is visualized in Figure 9.

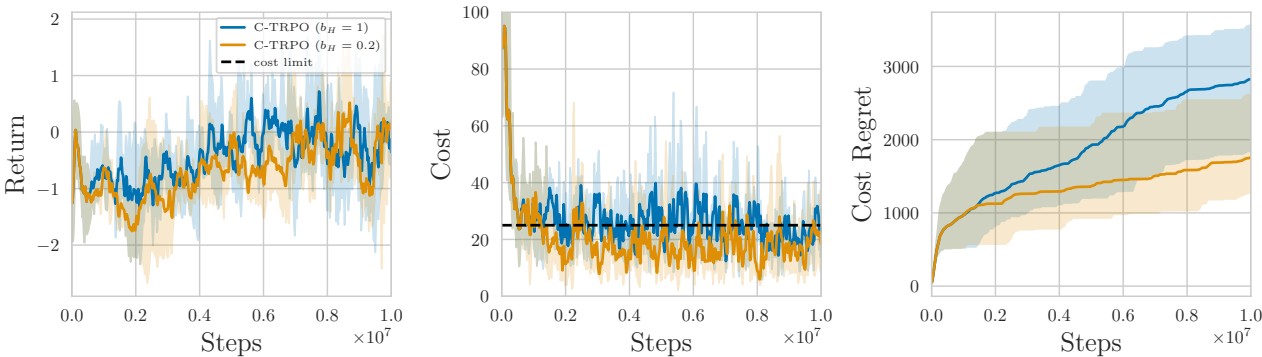

Figure 9: Influence of the hysteresis fraction $b_\mathrm{H}$ on C-TRPO's performance.

## D.2. Ablation Study: CPO vs. C-TRPO

We conduct an ablation study to rule out that our improvements of C-TRPO over CPO are only due to hysteresis. For this, we run both CPO and C-TRPO with and without hysteresis with the same hysteresis parameter as in our other experiments. We see that the hysteresis improves safety for both algorithm. Further, we find that the hysteresis slightly reduces the return of C-TRPO. Overall, we clearly see that C-TRPO itself is much safer compared to CPO as even C-TRPO without hysteresis achieves lower cost regret compared to CPO with hysteresis.

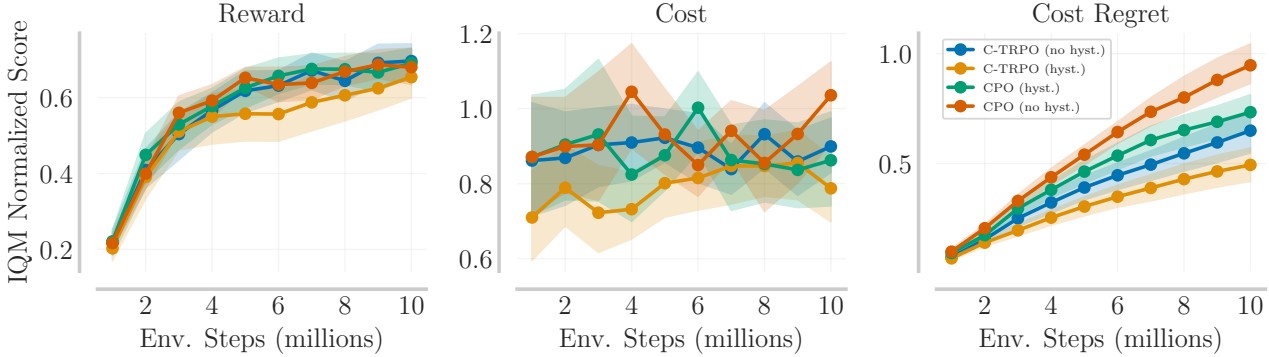

Figure 10: Ablation study on the core components of C-TRPO: Safe trust region (C-TRPO no hyst.) and recovery with hysteresis (CPO hyst.). Evaluation is based on the Inter Quartile Mean (IQM) normalized scores across 5 seeds and 8 tasks. From left to right: episode return of the reward (PPO normalized), episode return of the cost (threshold normalized), and cumulative cost violation (CPO normalized).

## D.3. Noisy Cost Estimates

To evaluate the sensitivity of C-TRPO to noisy or inaccurate estimates of the cost value function $V_c$, we train multiple policies with C-TRPO on the *AntVelocity* task and corrupt each policy's value estimate by varying levels of noise, i.e. white noise with varying standard deviations $\sigma$, see Figure 11.

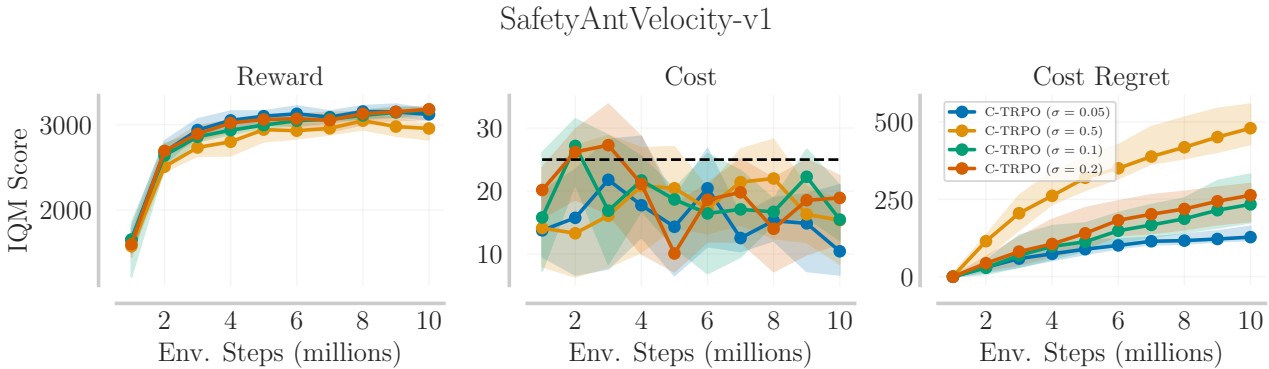

Figure 11: Performance of C-TRPO on the *AntVelocity* task as a function of cost value noise. The final reward decreases slightly for noise with $\sigma = 0.5$ and the cost regret increases with the amount of noise added to the value estimates.

## D.4. Performance on individual environments

Here, we compare C-TRPO to relevant baseline algorithms on all individual environments in terms of their sample efficiency curves. To improve readability of the plots, only the algorithms that are, on average, safe in the last iterate are included.

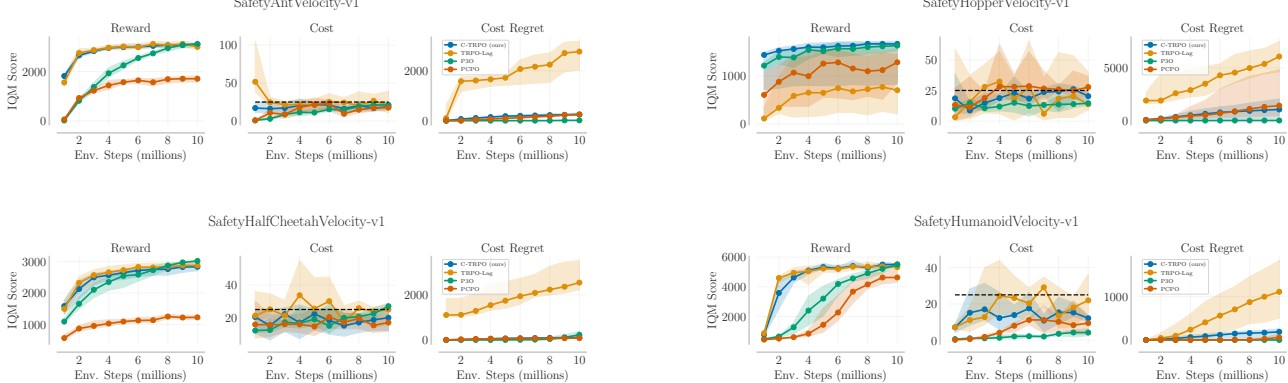

Figure 12: Benchmark on the locomotion environments.

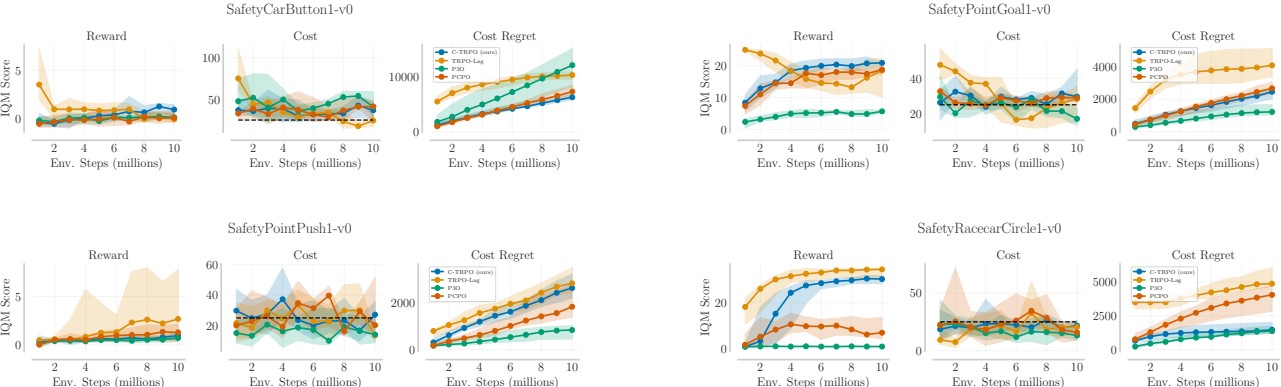

Figure 13: Benchmark on the navigation environments.

Table 1: Final performance per task for 5 seeds each, where expected reward $V_r$, expected cost $V_c$, and cost regret $Reg_+$ are shown, including lower and upper confidence intervals. We highlight the best performance with respect to the IQM of the return $V_r$ in bold, and box the lowest cost regret $Reg_+$.

| | | AntVelocity | HalfCheetahVelocity | HumanoidVelocity | HopperVelocity | CarButton | PointGoal | RacecarCircle | PointPush |
|---|---|---|---|---|---|---|---|---|---|
| **C-TRPO (OURS)** | $V_r$ | 3099.0±-38.5/46.5 | 2833.8±-123.6/70.9 | 5513.1±-66.5/53.8 | **1669.2±-839.8/55.2** | 1.0±-0.7/0.5 | 20.9±-0.7/0.4 | 30.3±-2.1/1.4 | 0.9±-0.3/0.4 |
| | $V_c$ | 18.8±-3.2/6.5 | 20.0±-8.1/5.7 | 12.2±-2.1/0.9 | 20.4±-3.2/2.9 | 36.8±-9.9/9.9 | 29.6±-3.2/15.4 | 21.7±-2.3/4.7 | 27.1±-8.3/8.5 |
| | $Reg_+$ | 244.0±-24.6/60.5 | 105.3±-30.6/25.2 | 185.9±-109.0/72.8 | 1056.4±-804.5/1014.9 | 2459.6±-442.6/230.9 | 1465.8±-265.7/552.4 | 2626.2±-436.0/591.3 | |
| **TRPO-LAG** | $V_r$ | 3001.9±-86.5/177.8 | 2879.0±-64.6/104.9 | 5326.9±-137.0/173.1 | 700.8±-493.5/868.2 | **-0.0±-0.5/0.6** | 18.5±-8.4/3.8 | **34.6±-3.9/0.6** | **2.7±-2.4/5.0** |
| | $V_c$ | 20.8±-10.0/18.4 | 25.0±-9.1/3.9 | 22.0±-7.0/14.3 | 13.7±-4.5/26.5 | 24.4±-7.1/5.7 | 28.4±-13.5/6.8 | 20.6±-5.0/4.4 | 13.7±-3.9/3.2 |
| | $Reg_+$ | 2773.5±-738.5/414.3 | 2546.5±-318.1/977.8 | 1110.4±-607.6/719.7 | 10343.1±-734.2/624.4 | 4094.7±-727.7/1029.6 | 4889.9±-202.4/1164.8 | 2828.3±-408.5/653.6 | |
| **CPPO-PID** | $V_r$ | 3205.3±-186.5/76.7 | 3036.1±-36.7/10.7 | 5877.3±-111.4/84.8 | 1657.5±-65.5/61.0 | -1.2±-0.5/0.6 | 6.1±-3.0/4.8 | 8.1±-5.5/4.3 | 1.0±-0.6/1.1 |
| | $V_c$ | 26.2±-5.3/4.4 | 26.5±-2.7/7.2 | 20.3±-8.6/6.0 | 18.6±-9.0/8.1 | 23.8±-8.4/6.0 | 21.8±-4.4/6.8 | 33.3±-6.5/5.9 | 22.8±-11.1/9.9 |
| | $Reg_+$ | 1416.8±-201.6/328.3 | 2094.1±-351.0/417.9 | 913.9±-228.4/304.9 | 3649.6±-1018.2/695.1 | 2233.6±-274.0/681.0 | 2573.5±-317.7/897.3 | 1981.3±-237.1/293.2 | |
| **P3O** | $V_r$ | **3122.5±-111.4/24.6** | 3020.3±-44.8/12.8 | 5492.1±-45.0/118.7 | 1633.5±-107.7/49.0 | 0.2±-0.2/0.3 | 5.7±-0.5/0.3 | 0.9±-0.1/0.1 | 0.7±-0.4/0.6 |
| | $V_c$ | 21.2±-2.2/2.5 | 27.0±-2.4/1.1 | 4.2±-1.7/2.2 | 14.6±-1.7/1.6 | 40.9±-10.4/18.2 | 17.1±-4.0/6.2 | 13.1±-4.1/4.6 | 14.1±-5.4/9.4 |
| | $Reg_+$ | [11.0±-5.0/8.1] | 228.1±-128.5/146.6 | [0.0±-0.0/0.0] | 12133.6±-4026.3/3007.5 | [3256.0±-488.9/211.4] | 2247.0±-415.5/615.1 | | |
| **PCPO** | $V_r$ | 1709.8±-135.8/124.9 | 1228.3±-67.0/67.7 | 4632.0±-331.6/574.4 | 1285.3±-432.2/166.1 | 0.1±-0.3/0.7 | 18.8±-1.5/1.1 | 6.9±-2.7/6.7 | 1.3±-0.9/0.9 |
| | $V_c$ | 17.7±-4.1/1.2 | 17.0±-5.2/6.3 | 9.4±-6.1/4.3 | 14.6±-1.7/1.6 | 41.4±-17.2/6.4 | 28.9±-0.8/1.4 | 15.8±-6.6/8.0 | 20.3±-6.6/31.0 |
| | $Reg_+$ | 250.8±-16.8/52.9 | [81.2±-20.1/31.1] | 55.7±-34.7/79.4 | [13.6±-13.5/131.7] | 2673.7±-343.7/387.2 | [1367.9±-159.8/368.4] | [832.1±-388.7/186.9] | |
| **CPO** | $V_r$ | 3106.7±-92.4/21.5 | 2824.1±-97.7/104.2 | 5569.6±-248.7/349.3 | 1696.4±-16.5/19.4 | 1.1±-0.6/0.2 | 20.4±-0.8/2.0 | 0.7±-0.3/2.9 | 0.7±-0.3/2.9 |
| | $V_c$ | 25.1±-17.5/11.3 | 23.1±-16.8/8.0 | 16.2±-3.7/8.6 | 25.7±-8.1/4.4 | 33.5±-10.6/8.7 | 28.2±-3.4/4.1 | 23.1±-9.9/4.5 | 28.9±-8.0/20.0 |
| | $Reg_+$ | 1323.7±-195.1/284.9 | 1142.2±-319.2/209.0 | 116.9±-73.5/249.9 | 6067.1±-546.1/977.0 | 2532.2±-276.8/206.1 | 4059.8±-1219.8/690.6 | 1824.5±-449.2/968.5 | |
| **CUP** | $V_r$ | 3092.0±-53.8/170.7 | 2916.7±-346.9/116.8 | 5677.9±-173.1/40.2 | 1639.8±-88.2/63.1 | 2.3±-1.8/3.9 | 22.4±-7.7/1.2 | 0.5±-0.3/1.1 | 0.5±-0.3/1.1 |
| | $V_c$ | 25.1±-1.9/2.7 | 40.7±-27.7/34.1 | 24.0±-7.0/10.5 | 23.9±-8.6/14.7 | 71.5±-28.2/57.0 | 45.2±-5.4/4.1 | 21.6±-5.9/6.3 | 35.6±-11.0/14.8 |
| | $Reg_+$ | 3160.9±-377.1/641.8 | 4719.9±-915.0/3503.3 | 5091.0±-276.6/306.5 | 3568.0±-735.9/1988.6 | 11715.5±-2047.7/1636.0 | 4774.5±-484.2/2431.3 | 2701.4±-388.8/1840.8 | 3209.9±-1255.9/1967.1 |
| **FOCOPS** | $V_r$ | 2942.2±-96.3/45.6 | 2997.8±-380.0/17.8 | 5420.0±-262.0/183.0 | 1670.6±-19.4/8.3 | 1.9±-0.7/1.1 | 17.6±-3.1/3.6 | 0.7±-0.5/2.1 | 0.7±-0.5/2.1 |
| | $V_c$ | 28.1±-8.1/1.4 | 36.9±-7.5/3.2 | 10.9±-4.1/8.5 | 25.9±-2.4/5.5 | 33.0±-10.0/30.4 | 33.7±-10.8/20.4 | 24.7±-8.1/4.7 | 24.7±-8.4/4.8 |
| | $Reg_+$ | 2257.8±-437.4/248.0 | 872.7±-766.6/684.7 | 1299.1±-320.5/455.9 | 23726.4±-10827.1/20476.7 | 5917.6±-1987.9/7276.3 | 6031.9±-1427.1/1075.8 | 2684.0±-451.8/484.2 | |
| **PPO-LAG** | $V_r$ | 3210.7±-126.6/85.8 | **3035.6±-27.6/15.5** | 5814.9±-102.9/122.0 | 240.1±-92.7/159.0 | 0.3±-1.0/0.8 | **9.4±-1.3/1.8** | 0.6±-0.2/0.0 | 0.6±-0.2/0.0 |
| | $V_c$ | 28.9±-8.6/8.7 | 23.2±-2.8/1.9 | 12.7±-7.6/31.0 | 38.8±-24.4/36.4 | 39.2±-12.7/41.1 | 22.5±-4.3/10.1 | 31.7±-9.2/2.7 | 18.2±-11.4/9.5 |
| | $Reg_+$ | 1767.5±-224.5/194.1 | 3339.6±-486.9/512.1 | 872.7±-766.6/684.7 | 5909.8±-3420.8/2790.5 | 22554.4±-6386.9/10174.9 | 6322.2±-419.5/722.3 | 3464.7±-419.5/1743.8 | |
| **IPO** | $V_r$ | 2962.4±-39.2/31.6 | 2810.9±-143.1/124.8 | 5909.8±-320.5/455.9 | 1535.2±-857.1/167.6 | -0.5±-0.2/0.5 | 6.6±-3.3/3.0 | 1.5±-0.3/0.4 | 0.6±-0.2/0.3 |
| | $V_c$ | 28.3±-6.1/4.8 | 27.4±-3.5/7.9 | 18.1±-6.9/8.2 | 24.9±-10.3/2.0 | 38.5±-6.7/3.9 | 25.6±-2.0/6.5 | 24.6±-4.2/9.9 | 23.7±-9.0/3.9 |
| | $Reg_+$ | 1548.1±-469.7/459.5 | 2351.2±-1385.0/1098.6 | 580.3±-319.9/167.0 | 3958.3±-2464.1/2771.1 | 6570.8±-871.6/1370.8 | 5926.5±-938.8/1361.4 | 2464.2±-711.4/1688.5 | 3496.1±-717.2/1900.7 |

