# OpenReview forum: "Embedding Safety into RL: A New Take on Trust Region Methods"
_ICML.cc/2025/Conference — ICML 2025 poster_

### Official Review · Reviewer_seAB · 2025-03-13

**Overall Recommendation:** 4

**Summary:**

This paper considers the problem of constrained MDPs. The key idea is to modify the geometry of the policy space to ensure that trust regions contain only safe policy space. This is achieved by introducing a new family of policy divergence that incorporate certain mirror functions. The author provide theoretical guarantees on the convergence of CNPG and theoretical properties of the C-TRPO updates. In addtion, the author also present empirical evaluations across 8 different tasks to show that C-TRPO reduces constraint violations while maintaining competitive returns.

**Claims And Evidence:**

Yes. The theoretical properties (convergence and optimality) are well-supported by the proofs in section 4 and appendix C.
The empirical results demonstrate reduced contraint violations. The comparison is comprehensive and fair.

**Essential References Not Discussed:**

No

**Experimental Designs Or Analyses:**

8 tasks from safety gymnasium benchmark are included in the experiment for empirical analysis. Overall, the experimental design is sound.

**Methods And Evaluation Criteria:**

Yes, the proposed methods C-TRPO makes sense as it provides a theoretically sound way to incorporate safety constraints into the policy geometry. The evaluation framework comprehensively assesses both theoretical claims and practical performance while maintaining relevance to real-world safety constraints.

**Other Comments Or Suggestions:**

No

**Other Strengths And Weaknesses:**

Strengths:
1. the idea of combines trust region with barrier functions is novel, and it provides better solution for solving constrained MDPs.
2. This paper provides strong theoretical guarantees and clean math framework connecting trust regions and safety constraints.
Weaknesses:
1. The paper has limited evaluation on larger-scale problems.

**Questions For Authors:**

1. How sensitive is C-TRPO to the accuracy of the estimated cost function? Could you provide an analysis or experiments showing how performance degrades with increasingly noisy or misspecified constraints?
2. How does C-TRPO handle multiple, potentially conflicting constraints? Could you provide theoretical insights or empirical results for scenarios with multiple safety criteria?

**Relation To Broader Scientific Literature:**

This paper synthesizes ideas from multiple areas, applying concepts from optimization and control theory to address practical challenges in safe reinforcement learning. It represents a step forward in making RL more applicable to safety-critical real-world scenarios.

**Theoretical Claims:**

Yes, I examined several key theoretical proofs in the paper, particularly in Section 4 and Appendix C.I did not find any obvious errors in the proofs, though some proofs in the appendix (particularly for technical lemmas) could benefit from more detailed explanations of intermediate steps.

---

> ### Author Rebuttal · Authors · 2025-03-31
>
> > How sensitive is C-TRPO to the accuracy of the estimated cost function? Could you provide an analysis or experiments showing how performance degrades with increasingly noisy or misspecified constraints?
>
> This is an excellent question. While we do not yet have a comprehensive theoretical analysis, we are running experiments to assess C-TRPO’s robustness to noisy or misspecified cost function estimates and will report results in the updated appendix.
>
> > How does C-TRPO handle multiple, potentially conflicting constraints? Could you provide theoretical insights or empirical results for scenarios with multiple safety criteria?
>
> C-TRPO naturally extends to multiple constraints. The benchmark environments we consider already include multiple cost signals, which are aggregated into a single constrained objective. While our current experiments adhere to this setup, it is possible to modify the benchmark tasks to enforce separate constraints on individual cost functions.
>
> We will expand our discussion on handling multiple constraints in the final version of the paper. Theorem 4.5 already implies that the optimal constrained policy $\pi^*_{safe}$ found by C-NPG satisfies as few constraints with equality as required to be optimal, which is discussed in the respective paragraph. We expect similar results to hold for C-TRPO as well.
>
> Additionally, we recognize the importance of empirical evaluation in scenarios with distinct, potentially conflicting constraints, and we consider this an important direction for future work.

---

### Official Review · Reviewer_D8iN · 2025-03-14

**Overall Recommendation:** 4

**Summary:**

In this paper, authors present idea of solving Constrained Markov Decision Processes (CMDPs) using trust regions that obey the constraints strictly and allow for return maximization. Earlier approaches work with KL-divergence based trust regions and try to recover the policy if the constraints of CMDP are violated using hysteresis. The paper works in the state distribution space, where KL divergence is expressed as Bregman divergence and the constraints of the MDP are converted to barrier functions that make the divergence go infinitely high for violating the constraint. This creates a "safe" trust region under which a policy would be optimized using natural policy gradient-style updates. If due to empirical approximations, the constraint gets violated, the hysteresis approach would be used to bring policy back into safe region. The paper presents important theoretical results for designing safe trust region, guarantees for reaching the optimal policy for the given CMDP and safety guarantees during training. The methods proposed are compared against other baselines in safe RL literature on safety gym environments. The results demonstrate that the proposed approach provides high returns while keeping the violations minimal.

**Claims And Evidence:**

The contributions claimed in the work are supported with theoretical and empirical evidence.

**Essential References Not Discussed:**

None that I am aware of.

**Experimental Designs Or Analyses:**

- I had a problem understanding the results depicted in figure 3. Why are the results aggregated together across multiple tasks? How can we compare the average of normalized rewards and constraint violations across 8 tasks together?

**Methods And Evaluation Criteria:**

- The algorithm proposed in the paper is closely related to Trust Region Policy Optimization (TRPO) [1] and Constrained Policy Optimization (CPO) [2]. The components added to TRPO and CPO are justified properly and make sense.

- The paper uses safety gymnasium [3] for their experimentation. The choice of environments is apt. As for metrics, expected returns and constraint violations are compared across multiple algorithms.

[1] Schulman, J., Levine, S., Moritz, P., Jordan, M. I., and Abbeel, P. Trust region policy optimization, 2017

[2] Achiam, J., Held, D., Tamar, A., and Abbeel, P. Constrained policy optimization, 2017.

[3] Ji, J., Zhang, B., Zhou, J., Pan, X., Huang, W., Sun, R., Geng, Y., Zhong, Y., Dai, J., and Yang, Y. Safety gymnasium: A unified safe reinforcement learning benchmark. In Thirty-seventh Conference on Neural Information Processing Systems Datasets and Benchmarks Track, 2023

**Other Comments Or Suggestions:**

I had a few suggestions regarding writing:
- On line 233 (left column) the paper starts talking about divergence $D_C$ which is said to be defined below and there is no definition until it has been used multiple times before it is formally defined in equations 16 and 17.
- Also, algorithm 1 can be moved after the description of entire methodology that way it avoids unnecessary introduction of variables and steps that are not talked about in the text until algorithm 1 starts in the text.

**Other Strengths And Weaknesses:**

Strength of the paper is in its theoretical rigor and innovation behind design of safe trust region. The paper takes more principled route of adhering to the safe trust region while updating policies.

Weakness: I don’t see any major weakness of this work, although I have not checked the proofs in this work which can have mistakes.

**Questions For Authors:**

Question regarding Theorem 4.4, Isn't CPO also safety invariant? It would be sufficient to show the CPO is invariant and then, C-NPG with any choice of beta would be invariant too. I feel the property that needs to be more emphasized in the text is the conservativeness of the policy update under higher beta values w.r.t. CPO.

**Relation To Broader Scientific Literature:**

Unlike earlier methods, the paper presents a way to design a safe trust region and provides a practical algorithm to optimize for returns within this trust region. The barrier functions over state distribution is a clever way of inducing such a safe trust region.

The paper provides C-NPG, a natural policy gradient variation with safe trust region, and C-TRPO, TRPO under safe trust region, algorithms which naturally inherit the methodology from earlier prominent works with CPO's hysteresis idea that allows for safe policy updates.

**Theoretical Claims:**

I have not checked the proofs of propositions 4.1, 4.2 and 4.3 and the proofs of theorems 4.4 and 4.5.

---

> ### Author Rebuttal · Authors · 2025-03-31
>
> Thank you for your detailed comments and suggestions! We respond to each of your points in detail below.
>
> > I had a problem understanding the results depicted in figure 3. Why are the results aggregated together across multiple tasks? How can we compare the average of normalized rewards and constraint violations across 8 tasks together?
>
> We follow a standard procedure from unconstrained RL to provide a concise summary of results, adopting the methodology from [rliable](https://github.com/google-research/rliable):
>
> - Each algorithm is trained across multiple seeds per environment.
>
> - Performance is normalized per environment relative to a reference algorithm.
>
> - Normalized results are then aggregated across seeds and tasks to enable a high-level comparison.
>
> To ensure robustness, we use interquartile mean aggregation and report bootstrapped confidence intervals, following best practices in RL evaluation. While this approach provides an informative summary, we acknowledge that individual task-level variations are important, which is why we also include per-task sample efficiency curves in the appendix.
>
> > The barrier function over state distributions is a clever way of inducing such a safe trust region.
>
> Thank you for the positive assessment!
>
> > C-TRPO inherits methodology from CPO’s hysteresis idea that allows for safe policy updates.
>
> We would like to clarify that hysteresis was introduced in our work, not in the original CPO paper. However, our results show that applying hysteresis to CPO improves its performance, as reported in our ablation studies. To achieve the full performance, C-TRPO’s safe trust region is necessary.
>
> > On line 233 (left column), the paper discusses divergence before it is formally defined (Equations 16 and 17).
>
> Thank you for this suggestion! There are two key divergences:
> - The theoretical divergence $D_C$ (Equation 10).
> - The approximated divergence $\bar{D}_C$ (Equation 16).
>
> We will clarify this distinction earlier in the manuscript.
>
> > Algorithm 1 should be moved after the full methodology description to improve readability.
>
> We agree and will restructure the manuscript accordingly.
>
> > Question regarding Theorem 4.4, Isn't CPO also safety invariant? It would be sufficient to show the CPO is invariant and then, C-NPG with any choice of beta would be invariant too.
>
> Neither CPO nor C-TRPO are strictly safety-invariant as defined in Theorem 4.4 due to approximation and estimation errors. However, Proposition B.2 shows that in the limit of small step sizes, C-TRPO converges to the safety-invariant C-NPG.
>
> In contrast, CPO does not follow a natural gradient update, making its small-stepsize limit unclear. While a bound similar to Proposition 4.3 exists for CPO, it is less conservative than C-TRPO’s. Importantly, both methods provide only a bounded approximation of the ideal safety invariance property of C-NPG, even with perfect value function knowledge. When value functions are estimated from finite samples, these bounds are further affected.
>
> In practice, C-TRPO offers a more conservative safety guarantee with minimal computational overhead or performance loss. Since estimation errors also impact safety, we plan to explore the finite-sample safety properties of both methods in future work.
>
> > I feel the property that needs to be more emphasized in the text is the conservativeness of the policy update under higher beta values w.r.t. CPO.
>
> Thank you for this suggestion. We will highlight this aspect in the manuscript.

---

> > ### Comment · Reviewer_D8iN · 2025-04-07
> >
> > In lieu of authors addressing most of my concerns and other reviewers having checked the theoretical correctness, I am increasing my score to 4 (accept).

---

### Official Review · Reviewer_kedz · 2025-03-14

**Overall Recommendation:** 4

**Summary:**

The paper introduces Constrained Trust Region Policy Optimization, a new Constrained RL algorithm based Trust Region Policy methods. The trust region is made to only contain safe policies for the update step. The new algorithm enjoys good theoretical properties, with improvement and safety guarantees similar to Constrained Policy Optimization while having provable safety and convergence properties with some additional assumptions. Because of constraining the trust region to safe policies only, the algorithm is designed incur less constraint violations during training than the popular lagrangian approaches. The algorithm is benchmarked on different experiments, where it is shown to be competitive with state-of-the-art algorithms like CPO or CUP.

**Claims And Evidence:**

The practical claims are that the algorithm is competitive with state-of-the-art algorithms, with a focus on the number of safety violations during training while still providing good optimal reward. The claim is supported by the experiments where the algorithm is benchmarked against other state-of-the-art algorithms in Safety Gym environments.

**Essential References Not Discussed:**

I do not see any essential references that are not discussed in the paper.

**Experimental Designs Or Analyses:**

The experimental designs are evaluating the influence of the hyperparameters, comparing C-TRPO to CPO with hysteresis, and more generally evaluating C-TRPO versus other state-of-the-art algorithms on Safety Gym environments

**Methods And Evaluation Criteria:**

The method for evaluating the performance of the algorithm is benchmarking the algorithm on Safety Gym environments, and the criteria for the evaluation of the algorithm is the number of violations during training, and the cost and average reward at the end of training.

**Other Comments Or Suggestions:**

The paper provides good background and intuition, and the subtleties of the links with NPG are well-explained.

**Other Strengths And Weaknesses:**

The paper's strength is designing a new approach that has very good theoretical properties that are shown (maybe a regret analysis would strengthen the paper even more). The experiments are also quite convincing: even though the improvement in practice is not extreme, it is still significant. The paper is well-written.

**Questions For Authors:**

I have no questions for the authors.

**Relation To Broader Scientific Literature:**

C-TRPO fits into a growing body of primal and primal-dual methods for solving constrained MDPs, and is a primal approach closely tied to CPO and TRPO. To the best of my knowledge, the approach in the paper is new and different from e.g. PCPO.

**Theoretical Claims:**

The theoretical claims are mainly Proposition 4.1 (reward improvement) and Proposition 4.3 (worst-case constraint violation), Theorem 4.4 (safety during training) and Theorem 4.5 (convergence).

---

> ### Author Rebuttal · Authors · 2025-03-31
>
> Thank you for your positive assessment of our work!
>
> We agree that a regret analysis would further strengthen the theoretical contribution and see this as a valuable direction for follow-up work.

---

### Official Review · Reviewer_gPum · 2025-03-14

**Overall Recommendation:** 4

**Summary:**

The paper proposes C-TRPO and C-NPG for solving CMDPs. Mirror functions are used to define policy divergences that are finite only for safe policies. This divergence metric is then used to reshape the policy shape geometry to ensure that trust regions contain only safe policies. The algorithms are analysed theoretically, and it is empirically shown that they lead to fewer constraint violations while obtaining similar rewards.

**Claims And Evidence:**

The claims in the paper are clear and convincing.

**Essential References Not Discussed:**

From my knowledge, I think the paper covers the essential references well.

**Experimental Designs Or Analyses:**

The experiments are evaluated on a well-established safe RL benchmark and are sound, in my opinion.

**Methods And Evaluation Criteria:**

The experiments are thorough and the benchmark considered is widely used for safe RL methods. I wonder why log-barrier-based approaches such as (1 or 2) are not used as baselines. Furthermore, I would have also expected Saute RL (3) to be a baseline.


1. https://www.jmlr.org/papers/volume25/22-0878/22-0878.pdf
2. https://arxiv.org/abs/2410.09486
3. https://arxiv.org/pdf/2202.06558

**Other Comments Or Suggestions:**

No additional comments.

**Other Strengths And Weaknesses:**

**Strengths**:

-- The paper tackles an important problem of safe RL using widely applied trust region approaches.

-- The paper is well written and easy to follow.

-- The experiments are thorough.

**Weaknesses**:

-- Log barrier baselines are not considered in the experiment.

-- I am not sure about analysing cost-regret with trust region methods given that they are rarely applied for learning in the real world due to sample inefficiency.

**Questions For Authors:**

See strengths and weaknesses.

**Relation To Broader Scientific Literature:**

The problem addressed is important and relevant. The insights drawn for trust region methods are also broadly important. The only thing I am unsure about is that trust region methods are generally very sample-inefficient and, therefore, rarely applied for direct learning in the real world. The common strategy is to train the policy in simulation. In this case, what is the benefit of analysing the cost-regret? Effectively, what we care about is that the final policy is safe. I would appreciate it if the authors could discuss this in the paper.

**Theoretical Claims:**

I think the theoretical claims are correct.

---

> ### Author Rebuttal · Authors · 2025-03-31
>
> Thank you for your valuable feedback! We have addressed your specific comments in detail below.
>
> > Log barrier baselines are not considered in the experiment.
>
> We considered **IPO** as a practical log-barrier baseline, along with **P3O**, a proximal adaptation.
>
> Regarding the specific works mentioned:
>
> - LB-SGD (Usmanova et al.) is discussed in our related work section. However, our focus is on practical deep RL algorithms, whereas LB-SGD primarily offers theoretical insights. Given the lack of large-scale empirical evaluations and implementation details for deep function approximation in LB-SGD, we did not include it as a baseline in our experiments.
>
> - Saute RL presents an interesting comparison but tackles a subtly different problem formulation than standard CMDPs. As seen in Definition 4 of their work, Saute RL imposes stricter constraints by disallowing stochastic policies from occasionally violating constraints and assuming non-negative safety costs. While this formulation is reasonable for safety, it does not always correspond to a general CMDP. A key strength of C-TRPO is its reliability in solving standard CMDPs, making it applicable to broader tasks like diverse policy optimization (e.g., [1]). A rigorous theoretical and empirical comparison would require additional work beyond the current scope. However, we will mention Saute RL in the manuscript and highlight its comparison with C-TRPO as an important direction for future research.
>
> - ActSafe is a newly accepted work that we were not previously aware of. Since it falls under model-based safe RL, which we briefly discuss in our related work section, we will add a reference in the final manuscript. However, we want to highlight key differences from our approach. Specifically, in Definition 4.7, ActSafe indeed defines a distance measure between policies, which is informed by cost continuity. In contrast, our Bregman divergence explicitly captures differences in the cost values of the policies, representing a key conceptual distinction in how safety is enforced.
>
> [1] Zahavy, Tom, et al. "Discovering Policies with DOMiNO: Diversity Optimization Maintaining Near Optimality." The Eleventh International Conference on Learning Representations.
>
> > I am not sure about analysing cost-regret with trust region methods given that they are rarely applied for learning in the real world due to sample inefficiency
>
> This is an important point. While near on-policy approaches like TRPO tend to be sample inefficient, they remain valuable due to their stability and well-established theoretical foundations in the function approximation setting. While they may not be ideal for online real-world learning, understanding how to minimize cumulative constraint violation regret is essential, as any algorithm used for fine-tuning on a real system must consider it.
>
> We believe that the insights from C-TRPO’s regret analysis will contribute to the development of future algorithms that are both sample-efficient and safety-aware.

---

### Decision · Program_Chairs · 2025-05-01

**Decision:**

Accept (poster)

**Comment:**

This paper introduces C-TRPO, a novel approach to safe reinforcement learning by reshaping policy space geometry so that trust regions inherently contain only safe policies. This theoretically grounded method extends trust region policy optimization with safety guarantees, ensuring constraint satisfaction throughout training without sacrificing return. Reviewers consistently praise the paper for its strong theoretical contributions, rigorous empirical evaluation across Safety Gym environments, and thoughtful handling of reviewer concerns. While some noted the lack of certain baselines and deeper empirical tests for constraint robustness, the consensus is that the work presents a significant step forward in safe RL, justifying its acceptance.